# Locally Private Gaussian Estimation

**Matthew Joseph** [*]
University of Pennsylvania
majos@cis.upenn.edu

**Janardhan Kulkarni**
Microsoft Research Redmond
jakul@microsoft.com

**Jieming Mao** [†]
Google Research New York
maojm@google.com

**Zhiwei Steven Wu** [‡]
University of Minnesota
zsw@umn.edu

## Abstract

We study a basic private estimation problem: each of $n$ users draws a single i.i.d. sample from an unknown Gaussian distribution $N(\mu, \sigma^2)$, and the goal is to estimate $\mu$ while guaranteeing *local differential privacy* for each user. As minimizing the number of rounds of interaction is important in the local setting, we provide *adaptive* two-round solutions and *nonadaptive* one-round solutions to this problem. We match these upper bounds with an information-theoretic lower bound showing that our accuracy guarantees are tight up to logarithmic factors for all sequentially interactive locally private protocols.

## 1 Introduction

Differential privacy is a formal algorithmic guarantee that no single input has a large effect on the output of a computation. Since its introduction [11], a rich line of work has made differential privacy a compelling privacy guarantee (see Dwork et al. [12] and Vadhan [24] for surveys), and deployments of differential privacy now exist at many organizations, including Apple [2], Google [5, 13], Microsoft [8], Mozilla [3], and the US Census Bureau [1, 20].

Much recent attention, including almost all industrial deployments, has focused on a variant called *local differential privacy* [4, 11, 19]. In the local model private data is distributed across many users, and each user privatizes their data *before* the data is collected by an analyst. Thus, as any locally differentially private computation runs on already-privatized data, data contributors need not worry about compromised data analysts or insecure communication channels. In contrast, (global) differential privacy assumes that the data analyst has secure, trusted access to the unprivatized data.

However, the stronger privacy guarantees of the local model come at a price. For many problems, a locally private solution requires far more samples than a globally private solution [7, 10, 19, 23]. Here, we study the basic problem of locally private Gaussian estimation: given $n$ users each holding an i.i.d. draw from an unknown Gaussian distribution $N(\mu, \sigma^2)$, can an analyst accurately estimate the mean $\mu$ while guaranteeing local differential privacy for each user?

On the technical front, locally private Gaussian estimation captures two general challenges in locally private learning. First, since data is drawn from a Gaussian, there is no a priori (worst-case) bound on the scale of the observations. Naive applications of standard privatization methods like Laplace and Gaussian mechanisms must add noise proportional to the worst-case scale of the data and are thus infeasible. Second, protocols requiring many rounds of user-analyst interaction are difficult to

---

[*] A portion of this work was done while at Microsoft Research Redmond.

[†] This work done while at the Warren Center, University of Pennsylvania.

[‡] A portion of this work was done while at Microsoft Research New York.

implement in real-world systems and may incur much longer running times. Network latency as well as server and user liveness constraints compound this difficulty [22]. It is therefore desirable to limit the number of *rounds of interaction* between users and the data analyst. Finally, besides being a fundamental learning problem, Gaussian estimation has several real-world applications (e.g. telemetry data analysis [8]) where one may assume that users' behavior follows a Guassian distribution.

## 1.1 Our Contributions

We divide our solution to locally private Gaussian estimation into two cases. In the first case, $\sigma$ is known to the analyst, and in the second case $\sigma$ is unknown but bounded in known $[\sigma_{\min}, \sigma_{\max}]$. For each case, we provide an $(\varepsilon, 0)$-locally private adaptive two-round protocol and nonadaptive one-round protocol[4]. Our privacy guarantees are worst-case; however, when $x_1, \ldots, x_n \sim N(\mu, \sigma^2)$ we also get the following accuracy guarantees.

**Theorem 1.1** (Informal). *When $\sigma$ is known, and $n$ is sufficiently large, there exists two-round protocol outputting $\hat{\mu}$ such that $|\hat{\mu} - \mu| = O\left(\frac{\sigma}{\varepsilon}\sqrt{\frac{\log(1/\beta)}{n}}\right)$ with probability $1 - \beta$, and there exists one-round protocol outputting $\hat{\mu}$ such that $|\hat{\mu} - \mu| = O\left(\frac{\sigma}{\varepsilon}\sqrt{\frac{\log(1/\beta)\sqrt{\log(n)}}{n}}\right)$ with probability $1 - \beta$.*

**Theorem 1.2** (Informal). *When $\sigma$ is unknown but bounded in known $[\sigma_{\min}, \sigma_{\max}]$, and $n$ is sufficiently large, there exists two-round protocol outputting $\hat{\mu}$ such that $|\hat{\mu} - \mu| = O\left(\frac{\sigma}{\varepsilon}\sqrt{\frac{\log(1/\beta)\log(n)}{n}}\right)$ with probability $1 - \beta$, and there exists one-round protocol outputting $\hat{\mu}$ such that $|\hat{\mu} - \mu| = O\left(\frac{\sigma}{\varepsilon}\sqrt{\frac{\log([\sigma_{\max}/\sigma_{\min}]+1)\log(1/\beta)\log^{3/2}(n)}{n}}\right)$ with probability $1 - \beta$.*

All of our protocols are *sequentially interactive* [10]: each user interacts with the protocol at most once. We match these upper bounds with a lower bound showing that our results are tight for all sequentially interactive locally private protocols up to logarithmic factors. We obtain this result by introducing tools from the *strong data processing inequality* literature [6, 21]. Using subsequent work by Joseph et al. [16], we can also extend this lower bound to fully interactive protocols.

**Theorem 1.3** (Informal). *For a given $\sigma$, there does not exist an $(\varepsilon, \delta)$-locally private protocol $\mathcal{A}$ such that for any $\mu = O\left(\frac{\sigma}{\varepsilon}\sqrt{\frac{1}{n}}\right)$, given $x_1, \ldots, x_n \sim N(\mu, \sigma^2)$, $\mathcal{A}$ outputs estimate $\hat{\mu}$ satisfying $|\hat{\mu} - \mu| = o\left(\frac{\sigma}{\varepsilon}\sqrt{\frac{1}{n}}\right)$ with probability $\geq 15/16$.*

## 1.2 Related Work

Several works have already studied differentially private versions of various statistical tasks, especially in the global setting. Karwa and Vadhan [18] and Kamath et al. [17] consider similar versions of Gaussian estimation under global differential privacy, respectively in the one-dimensional and high-dimensional cases. For both the known and unknown variance cases, Karwa and Vadhan [18] offer an $O\left(\sigma\sqrt{\frac{\log(1/\beta)}{n}} + \frac{\text{poly}\log(1/\beta)}{\varepsilon n}\right)$ accuracy upper bound for estimating $\mu$. Since an $\Omega\left(\sigma\sqrt{\frac{\log(1/\beta)}{n}}\right)$ accuracy lower bound holds even without privacy, our upper and lower bounds show that local privacy adds a roughly $\sqrt{n}$ accuracy cost over global privacy.

In concurrent independent work, Gaboardi et al. [14] also study locally private Gaussian estimation. We match or better their accuracy results with much lower round complexity. They provide adaptive protocols for the known- and unknown-$\sigma$ settings, with the latter protocol having round complexity $T$ as large as $\Omega(n)$, linear in the number of users. In contrast, we provide both adaptive and nonadaptive solutions, and our protocols all have round complexity $T \leq 2$. A full comparison appears in Figure 1.

Gaboardi et al. [14] also prove a tight lower bound for nonadaptive protocols that can be extended to sequentially interactive protocols. We provide a lower bound that is tight for sequentially interactive protocols up to logarithmic factors, and we depart from previous local privacy lower bounds by introducing tools from the *strong data processing inequality* (SDPI) literature [6, 21]. This approach

| Setting | Gaboardi et al. [14] Accuracy $\alpha$, Round Complexity $T$ | This Work Accuracy $\alpha$, Round Complexity $T$ |
|---|---|---|
| Known $\sigma$, adaptive | $\alpha = O\left(\frac{\sigma}{\varepsilon}\sqrt{\frac{\log\left(\frac{1}{\beta}\right)\log\left(\frac{n}{\beta}\right)\log\left(\frac{1}{\delta}\right)}{n}}\right)$ $T = 2$ | $\alpha = O\left(\frac{\sigma}{\varepsilon}\sqrt{\frac{\log\left(\frac{1}{\beta}\right)}{n}}\right)$ $T = 2$ |
| Known $\sigma$, nonadaptive | – | $\alpha = O\left(\frac{\sigma}{\varepsilon}\sqrt{\frac{\log\left(\frac{1}{\beta}\right)\sqrt{\log(n)}}{n}}\right)$ $T = 1$ |
| Unknown $\sigma$, adaptive | $\alpha = O\left(\frac{\sigma}{\varepsilon}\sqrt{\frac{\log\left(\frac{1}{\beta}\right)\log\left(\frac{n}{\beta}\right)\log\left(\frac{1}{\delta}\right)}{n}}\right)$ $T = \Omega\left(\log\left(\frac{R}{\sigma_{\min}}\right)\right)$ | $\alpha = O\left(\frac{\sigma}{\varepsilon}\sqrt{\frac{\log\left(\frac{1}{\beta}\right)\log(n)}{n}}\right)$ $T = 2$ |
| Unknown $\sigma$, nonadaptive | – | $\alpha = O\left(\frac{\sigma}{\varepsilon}\sqrt{\frac{\log\left(\frac{\sigma_{\max}}{\sigma_{\min}}+1\right)\log\left(\frac{1}{\beta}\right)\log^{3/2}(n)}{n}}\right)$ $T = 1$ |

Figure 1: A comparison of upper bounds in Gaboardi et al. [14] and here. In all cases, Gaboardi et al. [14] use $(\varepsilon, \delta)$-locally private algorithms and we use $(\varepsilon, 0)$. Here, $R$ denotes an upper bound on both $\mu$ and $\sigma$. In our setting, the upper bound on $\mu$ is $O(2^{n\varepsilon^2/\log(n/\beta)})$, leading the unknown variance protocol of Gaboardi et al. [14] to round complexity potentially as large as $\tilde{\Omega}(n\varepsilon^2/\log(1/\beta))$.

uses an SDPI to control how much information a sample gives about its generating distribution, then uses existing local privacy results to bound the mutual information between a sample and the privatized output from that sample. Subsequent work by Duchi and Rogers [9] generalizes the SDPI framework to prove lower bounds for a broader class of problems in local privacy. They also extend the SDPI framework to prove lower bounds for fully interactive algorithms.

## 2 Preliminaries

We consider a setting where, for each $i \in [n]$, user $i$'s datum is a single draw from an unknown Gaussian distribution, $x_i \sim N(\mu, \sigma^2)$, and these draws are i.i.d. In our communication protocol, users may exchange messages over public channels with a single (possibly untrusted) central analyst.[5] The analyst's task is to accurately estimate $\mu$ while guaranteeing local differential privacy for each user.

To minimize interaction with any single user, we restrict our attention to *sequentially interactive* protocols. In these protocols, every user sends at most a single message in the entire protocol. We also study the *round complexity* of these interactive protocols. Formally, one round of interaction in a protocol consists of the following two steps: 1) the analyst selects a subset of users $S \subseteq [n]$, along with a set of randomizers $\{Q_i \mid i \in S\}$, and 2) each user $i$ in $S$ publishes a message $y_i = Q_i(x_i)$.

A randomized algorithm is *differentially private* if arbitrarily changing a single input does not change the output distribution "too much". This preserves privacy because the output distribution is insensitive to any change of a single user's data. We study a stronger privacy guarantee called *local differential privacy*. In the local model, each user $i$ computes their message using a *local randomizer*. A local randomizer is a differentially private algorithm taking a single-element database as input.

**Definition 2.1** (Local Randomizer). *A randomized function $Q_i : X \to Y$ is an $(\varepsilon, \delta)$-local randomizer if, for every pair of observations $x_i, x_i' \in X$ and any $S \subseteq Y$, $\Pr[Q_i(x_i) \in S] \le e^\varepsilon \Pr[Q_i(x_i') \in S] + \delta$.*

A protocol is locally private if every user computes their message using a local randomizer. In a sequentially interactive protocol, the local randomizer for user $i$ may be chosen adaptively based on previous messages $z_1, \ldots, z_{i-1}$. However, the choice of randomizer cannot be based on user $i$'s data.

**Definition 2.2.** *A sequentially interactive protocol $\mathcal{A}$ is $(\varepsilon, \delta)$-locally private for private user data $\{x_1, \ldots, x_n\}$ if, for every user $i \in [n]$, the message $Y_i$ is computed using an $(\varepsilon, \delta)$-local randomizer $Q_i$. When $\delta > 0$, we say $\mathcal{A}$ is* approximately *locally private. If $\delta = 0$, $\mathcal{A}$ is* purely *locally private.*

## 3 Estimating $\mu$ with Known $\sigma$

We begin with the case where $\sigma^2$ is known (shorthanded "KV"). In Section 3.1, we provide a protocol KVGAUSSTIMATE that requires two rounds of analyst-user interaction. In Section 3.2, we provide a protocol 1ROUNDKVGAUSSTIMATE achieving a weaker accuracy guarantee in a single round. All omitted pseudocode and proofs appear in the full version of this paper [15].

### 3.1 Two-round Protocol KVGAUSSTIMATE

In KVGAUSSTIMATE the users are split into halves $U_1$ and $U_2$. In round one, the analyst queries users in $U_1$ to obtain an $O(\sigma)$-accurate estimate $\hat{\mu}_1$ of $\mu$. In round two, the analyst passes $\hat{\mu}_1$ to users in $U_2$, who respond based on $\hat{\mu}_1$ and their own data. The analyst then aggregates this second set of responses into a better final estimate of $\mu$.

**Theorem 3.1.** *Two-round protocol KVGAUSSTIMATE satisfies $(\varepsilon, 0)$-local differential privacy for $x_1, \ldots, x_n$ and, if $x_1, \ldots, x_n \sim_{iid} N(\mu, \sigma^2)$ where $\sigma$ is known and $\frac{n}{\log(n)} = \Omega\left(\frac{\log(\mu)\log(1/\beta)}{\varepsilon^2}\right)$, with probability $1 - \beta$ outputs $\hat{\mu}$ such that $|\hat{\mu} - \mu| = O\left(\frac{\sigma}{\varepsilon}\sqrt{\frac{\log(1/\beta)}{n}}\right)$.*

---

**Algorithm 1** KVGAUSSTIMATE

**Input:** $\varepsilon, k, \mathcal{L}, n, \sigma, U_1, U_2$
 1: **for** $j \in \mathcal{L}$ **do**
 2:     **for** user $i \in U_1^j$ **do**
 3:         User $i$ outputs $\tilde{y}_i \leftarrow \text{RR1}(\varepsilon, i, j)$
 4:     **end for**
 5: **end for**                                                   ▷ End of round 1
 6: Analyst computes $\hat{H}_1 \leftarrow \text{KVAGG1}(\varepsilon, k, \mathcal{L}, U_1)$
 7: Analyst computes $\hat{\mu}_1 \leftarrow \text{ESTMEAN}(\beta, \varepsilon, \hat{H}_1, k, \mathcal{L})$
 8: **for** user $i \in U_2$ **do**
 9:     User $i$ outputs $\tilde{y}_i \leftarrow \text{KVRR2}(\varepsilon, i, \hat{\mu}_1, \sigma)$
10: **end for**                                                 ▷ End of round 2
11: Analyst computes $\hat{H}_2 \leftarrow \text{KVAGG2}(\varepsilon, n/2, U_2)$
12: Analyst computes $\hat{T} \leftarrow \sqrt{2} \cdot \text{erf}^{-1}\left(\frac{2(-\hat{H}_2(-1)+\hat{H}_2(1))}{n}\right)$
13: Analyst outputs $\hat{\mu}_2 \leftarrow \sigma\hat{T} + \hat{\mu}_1$
**Output:** Analyst estimate $\hat{\mu}_2$ of $\mu$

---

#### 3.1.1 First round of KVGAUSSTIMATE

For neatness, let $L = \lfloor n/(2k)\rfloor$, $L_{\min} = \lfloor \log(\sigma)\rfloor$, $L_{\max} = L_{\min} - 1 + L$, and $\mathcal{L} = \{L_{\min}, L_{\min} + 1, \ldots, L_{\max}\}$. $U_1$ is then split into $L$ subgroups indexed by $\mathcal{L}$, and each subgroup has size $k = \Omega\left(\frac{\log(n/\beta)}{\varepsilon^2}\right)$. KVGAUSSTIMATE begins by iterating through each subgroup $j \in \mathcal{L}$. Each user $i \in U_1^j$ releases a privatized version of $\lfloor x_i/2^j\rfloor \bmod 4$ via randomized response (RR1): with probability $e^\varepsilon/(e^\varepsilon + 3)$, user $i$ outputs $\lfloor x_i/2^j\rfloor \bmod 4$, and otherwise outputs one of the remaining elements of $\{0, 1, 2, 3\}$ uniformly at random. Responses from group $U_1^j$ will be used to estimate the $j^{th}$ least significant bit of $\mu$ (rounded to an integer). The analyst then uses KVAGG1 ("Known Variance Aggregation") to aggregate and debias responses to account for this randomness.

---

**Algorithm 2** KVAGG1

---
**Input:** $\varepsilon, k, \mathcal{L}, U$
1: **for** $j \in \mathcal{L}$ **do**
2:    **for** $a \in \{0,1\}$ **do**
3:       $C^j(a) \leftarrow |\{\tilde{y}_i \mid i \in U^j, \tilde{y}_i = a\}|$
4:       $\hat{H}^j(a) \leftarrow \frac{e^\varepsilon+3}{e^\varepsilon-1} \cdot \left(C^j(a) - \frac{k}{e^\varepsilon+3}\right)$
5:    **end for**
6: **end for**
7: Output $\hat{H}$
**Output:** Aggregated histogram $\hat{H}$ of private user responses

---

The result is a collection of histograms $\hat{H}_1$. The analyst uses $\hat{H}_1$ in ESTMEAN to binary search for $\mu$. Intuitively, for each subgroup $U_1^j$ if all multiples of $2^j$ are far from $\mu$ then Gaussian concentration implies that almost all users $i \in U_1^j$ compute the same value of $\lfloor x/2^j \rfloor \bmod 4$. This produces a histogram $\hat{H}_1^j$ where most elements fall concentrate in a single bin. The analyst in turn narrows their search range for $\mu$. For example, if $\hat{H}_1^{L_{\max}}$ concentrates in 0, then the range narrows to $\mu \in [0, 2^{L_{\max}})$; if $\hat{H}_1^{L_{\max}-1}$ concentrates in 1, then the range narrows to $\mu \in [2^{L_{\max}-1}, 2^{L_{\max}})$, and so on.

If instead some multiple of $2^j$ is near $\mu$, the elements of $\hat{H}_1^j$ will spread over multiple (adjacent) bins. This is also useful: a point from the "middle" of this block of bins is $O(\sigma)$-close to $\mu$. The analyst thus takes such a point as $\hat{\mu}_1$ and ends their search. Our analysis will also rely on having a noticeably low-count bin that is non-adjacent to the bin containing $\mu$. This motivates using 4 as a modulus.

In this way, the analyst examines $\hat{H}_1^{L_{\max}}, \hat{H}_1^{L_{\max}-1}, \ldots$ in sequence, estimating $\mu$ from most to least significant bit. Crucially, the modulus structure of user responses enables the analyst to carry out this binary search with *one* round of interaction. Thus at the end of the first round the analyst obtains an $O(\sigma)$-accurate estimate $\hat{\mu}_1$ of $\mu$.

---

**Algorithm 3** ESTMEAN

---
**Input:** $\beta, \varepsilon, \hat{H}_1, k, \mathcal{L}$
1: $\psi \leftarrow \left(\frac{\varepsilon+4}{\varepsilon\sqrt{2}}\right) \cdot \sqrt{k \ln(8L/\beta)}$
2: $j \leftarrow L_{\max}$
3: $I_j \leftarrow [0, 2^{L_{\max}}]$
4: **while** $j \geq L_{\min}$ and $\max_{a \in \{0,1,2,3\}} \hat{H}_1^j(a) \geq 0.52k + \psi$ **do**
5:    Analyst computes integer $c$ such that $c2^j \in I_j$ and $c \equiv M_1(j) \bmod 4$
6:    Analyst computes $I_{j-1} \leftarrow [c2^j, (c+1)2^j]$
7:    $j \leftarrow j-1$
8: **end while**
9: $j \leftarrow \max(j, L_{\min})$
10: Analyst computes $M_1(j) \leftarrow \arg\max_{a \in \{0,1,2,3\}} \hat{H}_1^j(a)$
11: Analyst computes $M_2(j) \leftarrow \arg\max_{a \in \{0,1,2,3\}-\{M_1(j)\}} \hat{H}_1^j(a)$
12: Analyst computes $c^* \leftarrow$ maximum integer such that $c^*2^j \in I_j$ and $c^* \equiv M_1(j)$ or $M_2(j) \bmod 4$
13: Analyst outputs $\hat{\mu}_1 \leftarrow c^*2^j$
**Output:** Initial estimate $\hat{\mu}_1$ of $\mu$

---

### 3.1.2 Second round of KVGAUSSTIMATE

In the second round, the analyst passes $\hat{\mu}_1$ to users in $U_2$. Users respond through KVRR2 ("Known Variance Randomized Response"), a privatized version of an algorithm from the distributed statistical estimation literature [6]. In KVRR2, each user centers their point with $\hat{\mu}_1$, standardizes it using $\sigma$, and randomized responds on $\text{sgn}((x_i - \hat{\mu}_1)/\sigma)$. This crucially relies on the first estimate $\hat{\mu}_1$, as properly centering requires an initial $O(\sigma)$-accurate estimate of $\hat{\mu}$. The analyst can then aggregate these responses by a debiasing process KVAGG2 akin to KVAGG1.

---

**Algorithm 4** KVRR2

---

**Input:** $\varepsilon, i, \hat{\mu}_1, \sigma$
 1: User $i$ computes $x'_i \leftarrow (x_i - \hat{\mu}_1)/\sigma$
 2: User $i$ computes $y_i \leftarrow \mathrm{sgn}(x'_i)$
 3: User $i$ computes $c \sim_U [0,1]$
 4: **if** $c \le \frac{e^\varepsilon}{e^\varepsilon + 1}$ **then**
 5:     User $i$ publishes $\tilde{y}_i \leftarrow y_i$
 6: **else**
 7:     User $i$ publishes $\tilde{y}_i \leftarrow -y_i$
 8: **end if**
**Output:** Private centered user estimate $\tilde{y}_i$

---

**Algorithm 5** KVAGG2

---

**Input:** $\varepsilon, k, U$
 1: **for** $a \in \{-1, 1\}$ **do**
 2:     $C(a) \leftarrow |\{\tilde{y}_i \mid i \in U, \tilde{y}_i = a\}|$
 3:     $\hat{H}(a) \leftarrow \frac{e^\varepsilon + 1}{e^\varepsilon - 1} \cdot \left(C(a) - \frac{k}{e^\varepsilon + 1}\right)$
 4: **end for**
 5: Analyst outputs $\hat{H}$
**Output:** Aggregated histogram $\hat{H}$ of private user responses

---

From this aggregation $\hat{H}_2$, the analyst obtains a good estimate of the bias of the initial estimate $\hat{\mu}_1$. If $\hat{\mu}_1 < \mu$, responses will skew toward 1, and if $\hat{\mu}_1 > \mu$ responses will skew toward $-1$. By comparing this skew to the true standard CDF using the error function erf, the analyst recovers a better final estimate $\hat{\mu}_2$ of $\mu$ (Lines 12-13 of KVGAUSSTIMATE). Privacy of KVGAUSSTIMATE follows from the privacy of the randomized response mechanisms RR1 and KVRR2.

## 3.2 One-round Protocol 1ROUNDKVGAUSSTIMATE

Recall that in KVGAUSSTIMATE the analyst 1) employs user pool $U_1$ to compute rough estimate $\hat{\mu}_1$ and 2) adaptively refines this estimate using responses from the second user pool $U_2$. 1ROUNDKV-GAUSSTIMATE executes these two rounds of KVGAUSSTIMATE *simultaneously* by parallelization.

**Theorem 3.2.** *One-round protocol* 1ROUNDKVGAUSSTIMATE *satisfies* $(\varepsilon, 0)$*-local differential privacy for* $x_1, \ldots, x_n$ *and, if* $x_1, \ldots, x_n \sim_{iid} N(\mu, \sigma^2)$ *where* $\sigma$ *is known and* $\frac{n}{\log(n)} = \Omega\left(\frac{\log(\mu)\log(1/\beta)}{\varepsilon^2}\right)$, *with probability* $1 - \beta$ *outputs* $\hat{\mu}$ *such that*

$$|\hat{\mu} - \mu| = O\left(\frac{\sigma}{\varepsilon}\sqrt{\frac{\log(1/\beta)\sqrt{\log(n)}}{n}}\right).$$

1ROUNDKVGAUSSTIMATE splits $U_2$ into $\Theta(\sqrt{\log(n)})$ subgroups that run the second-round protocol from KVGAUSSTIMATE with different values of $\hat{\mu}_1$. Intuitively, it suffices that at least one subgroup centers using a $\hat{\mu}_1$ near $\mu$: the analyst can then use the data from that subgroup and discard the rest. By Gaussian concentration, most user samples cluster within $O(\sigma\sqrt{\log(n)})$ of $\mu$, so each subgroup $U_2^j$ receives a set of points $S(j)$ interspersed $\Theta(\sigma\sqrt{\log(n)})$ apart on the real line, and each user $i \in U_2^j$ centers using the point in $S(j)$ closest to $x_i$. This leads us to use $\Theta(\sqrt{\log(n)})$ groups with each point in $S(j+1)$ shifted $\Theta(\sigma)$ from the corresponding point in $S(j)$. By doing so, we ensure that some subgroup has most of its users center using a point within $O(\sigma)$ of $\mu$.

In summary, 1ROUNDKVGAUSSTIMATE works as follows: after collecting the single round of responses from $U_1$ and $U_2$, the analyst computes $\hat{\mu}_1$ using responses from $U_1$. By comparing $\hat{\mu}_1$ and $S(j)$ for each $j$, the analyst then selects the subgroup $U_2^{j^*}$ where most users centered using a value in $S(j^*)$ closest to $\hat{\mu}_1$. This mimics the effect of adaptively passing $\hat{\mu}_1$ to the users in $U_2^{j^*}$, so the analyst simply processes the responses from $U_2^{j^*}$ as it processed responses from $U_2$ in KVGAUSSTIMATE. Because $U_2^{j^*}$ contains $\Theta(n/\sqrt{\log(n)})$ users, the cost is a $\log^{1/4}(n)$ factor in accuracy.

# 4 Unknown Variance

In this section, we consider the more general problem with unknown variance $\sigma^2$ (shorthanded "UV") that lies in known interval $[\sigma_{\min}, \sigma_{\max}]$. We again provide a two-round protocol UVGAUSSTIMATE and a slightly less accurate one-round protocol 1ROUNDUVGAUSSTIMATE.

## 4.1 Two-round Protocol

UVGAUSSTIMATE is structurally similar to KVGAUSSTIMATE. In round one, the analyst uses the responses of half of the users to roughly estimate $\mu$, and in round two the analyst passes this estimate to the second half of users for improvement. However, two key differences now arise. First, since $\sigma$ is unknown, the analyst must now also estimate $\sigma$ in round one. Second, since the analyst does not have a very accurate estimate of $\sigma$, the refinement process of the second round employs Laplace noise rather than the CDF comparison used in KVGAUSSTIMATE.

**Theorem 4.1.** *Two-round protocol* UVGAUSSTIMATE *satisfies* $(\varepsilon, 0)$-*local differential privacy for* $x_1, \ldots, x_n$ *and, if* $x_1, \ldots, x_n \sim_{iid} N(\mu, \sigma^2)$ *where* $\sigma$ *is unknown but bounded in known* $[\sigma_{\min}, \sigma_{\max}]$ *and* $\frac{n}{\log(n)} = \Omega\left( \frac{\left[ \log\left( \frac{\sigma_{\max}}{\sigma_{\min}} + 1 \right) + \log(\mu) \right] \log\left( \frac{1}{\beta} \right)}{\varepsilon^2} \right)$, *with probability at least* $1 - \beta$ *outputs* $\hat{\mu}$ *such that*

$$|\hat{\mu} - \mu| = O\left( \frac{\sigma}{\varepsilon} \sqrt{\frac{\log(1/\beta)\log(n)}{n}} \right).$$

---

**Algorithm 6** UVGAUSSTIMATE

**Input:** $\varepsilon, k_1, \mathcal{L}_1, n, \sigma, U_1, U_2$
1: **for** $j \in \mathcal{L}_1$ **do**
2:     **for** user $i \in U_1^j$ **do**
3:         User $i$ outputs $\tilde{y}_i \leftarrow \text{RR1}(\varepsilon, i, j)$
4:     **end for**
5: **end for**                                                 ▷ End of round 1
6: Analyst computes $\hat{H}_1 \leftarrow \text{AGG1}(\varepsilon, \mathcal{L}_1, U_1)$
7: Analyst computes $\hat{\sigma} \leftarrow \text{ESTVAR}(\beta, \varepsilon, \hat{H}_1, k_1, \mathcal{L}_1)$
8: Analyst computes $\hat{H}_2 \leftarrow \text{KVAGG1}(\varepsilon, k_1, \mathcal{L}_1, U_1)$
9: Analyst computes $\hat{\mu}_1 \leftarrow \text{ESTMEAN}(\beta, \varepsilon, \hat{H}_2, k_1, \mathcal{L}_1)$
10: Analyst computes $I \leftarrow [\hat{\mu}_1 \pm \hat{\sigma}(2 + \sqrt{\ln(4n)})]$
11: **for** user $i \in U_2$ **do**
12:     User $i$ outputs $\tilde{y}_i \leftarrow \text{UVRR2}(\varepsilon, i, I)$
13: **end for**                                                ▷ End of round 2
14: Analyst outputs $\hat{\mu}_2 \leftarrow \frac{2}{n} \sum_{i \in U_2} \tilde{y}_i$
**Output:** Analyst estimate $\hat{\mu}_2$ of $\mu$

---

### 4.1.1 First round of UVGAUSSTIMATE

Similarly to KVGAUSSTIMATE, we split $U_1$ into $L_1 = \lfloor n/(2k_1) \rfloor$ subgroups of size $k_1 = \Omega\left( \frac{\log(n/\beta)}{\varepsilon^2} \right)$ and define $L_{\min} = \lfloor \log(\sigma_{\min}) \rfloor$, $L_{\max} = L_1 + L_{\min} - 1 \geq \lceil \log(\sigma_{\max}) \rceil$, and $\mathcal{L}_1 = \{L_{\min}, L_{\min} + 1, \ldots, L_{\max}\}$, indexing $U_1$ by $\mathcal{L}_1$.

Also as in KVGAUSSTIMATE, each user $i$ in each subgroup $U_1^j$ publishes a privatized version of $\lfloor x_i/2^j \rfloor \mod 4$. The analyst aggregates them (KVAGG1) into $\hat{H}_2$ and roughly estimates $\mu$ (ESTMEAN) as in KVGAUSSTIMATE. However, the analyst also employs a (similar) aggregation (AGG1) into $\hat{H}_1$ for estimating $\sigma$ (ESTVAR). At a high level, because samples from $N(\mu, \sigma^2)$ probably fall within $3\sigma$ of $\mu$, when $2^j \gg \sigma$ there exist $a, a + 1 \mod 4 \in \{0, 1, 2, 3\}$ such that almost all users $i$ have $\lfloor x_i/2^j \rfloor \mod 4 \in \{a, a + 1\}$. The analyst's debiased aggregated histogram $\hat{H}_1^j$ thus concentrates in at most two adjacent bins when $2^j \gg \sigma$ and spreads over more bins when $2^j \ll \sigma$. By a process like ESTMEAN, examining this transition from concentrated to unconcentrated in $\hat{H}_1^{L_{\max}}, \hat{H}_1^{L_{\max}-1}, \ldots$ yields a rough estimate of when $2^j \gg \sigma$ versus when $2^j \ll \sigma$. As a result, at the end of round one the analyst obtains $O(\sigma)$-accurate estimates $\hat{\sigma}$ of $\sigma$ and $\hat{\mu}_1$ of $\mu$.

#### 4.1.2 Second round of UVGAUSSTIMATE

The analyst now refines their initial estimate of $\mu$. First, the analyst constructs an interval $I$ of size $O(\hat{\sigma}\sqrt{\log(n)})$ around $\hat{\mu}_1$. Users in $U_2$ then truncate their values to $I$, add Laplace noise scaled to $|I|$ (the sensitivity of releasing a truncated point), and send the result to the analyst using UVRR2. The analyst then simply takes the mean of these responses as the final estimate of $\mu$. Its accuracy guarantee follows from concentration of user samples around $\mu$ and Laplace noise around 0. Privacy follows from our use of randomized response and Laplace noise.

We briefly explain our use of Laplace noise rather than CDF comparison. Roughly, when using an estimate $\hat{\sigma}$ in the centering process, error in $\hat{\sigma}$ propagates to error in the final estimate $\hat{\mu}_2$. This leads us to Laplace noise, which better handles the error in $\hat{\sigma}$ that estimation of $\sigma$ introduces. The cost is the $\sqrt{\log(n)}$ factor that arises from adding Laplace noise scaled to $|I|$. Our choice of $|I|$ — constructed to contain not only $\mu$ but the points of $\Omega(n)$ users — thus strikes a deliberate balance. $I$ is both large enough to cover most users (who would otherwise truncate too much and skew the responses) and small enough to not introduce much noise from privacy (as noise is scaled to $\mathsf{Lap}\left(|I|/\varepsilon\right)$).

### 4.2 One-round Protocol

We now provide a one-round version of UVGAUSSTIMATE, 1ROUNDUVGAUSSTIMATE.

**Theorem 4.2.** *One-round protocol* 1ROUNDUVGAUSSTIMATE *satisfies* $(\varepsilon, 0)$-*local differential privacy for* $x_1, \ldots, x_n$ *and, if* $x_1, \ldots, x_n \sim_{iid} N(\mu, \sigma^2)$ *where* $\sigma$ *is unknown but bounded in known* $[\sigma_{\min}, \sigma_{\max}]$ *and* $\frac{n}{\log(n)} = \Omega\left(\frac{\left[\log\left(\frac{\sigma_{\max}}{\sigma_{\min}}+1\right)+\log(\mu)\right]\log\left(\frac{1}{\beta}\right)}{\varepsilon^2}\right)$, *with probability at least* $1 - \beta$ *outputs* $\hat{\mu}$ *with*

$$|\hat{\mu} - \mu| = O\left(\frac{\sigma}{\varepsilon}\sqrt{\frac{\log\left(\frac{\sigma_{\max}}{\sigma_{\min}}+1\right)\log(1/\beta)\log^{3/2}(n)}{n}}\right).$$

Like 1ROUNDKVGAUSSTIMATE, 1ROUNDUVGAUSSTIMATE simulates the second round of UVGAUSSTIMATE simultaneously with its first round. 1ROUNDUVGAUSSTIMATE splits $U_2$ into subgroups, where each subgroup responds using a *different* interval $I_j$. At the end of the single round the analyst obtains estimates $\hat{\mu}_1$ and $\hat{\sigma}$ from users in $U_1$, constructs an interval $I$ from these estimates, and finds a subgroup of $U_2$ where most users employed a similar interval $I_j$. This similarity guarantees that the subgroup's responses yield the same accuracy as the two-round case up to an $O(\text{\# subgroups})$ factor. As in 1ROUNDKVGAUSSTIMATE, we rely on Gaussian concentration and the modulus trick to minimize the number of subgroups. However, this time we parallelize not only over possible values of $\hat{\mu}_1$ but possible values of $\hat{\sigma}$ as well. As this parallelization is somewhat involved, we defer its presentation to the full version of this paper [15].

In summary, at the end of the round the analyst computes $\hat{\mu}_1$ and $\hat{\sigma}$, computes the resulting interval $I^*$, and identifies a subgroup of $U_2$ that responded using an interval $I_j$ similar to $I^*$. This mimics the effect of passing an interval of size $O(\sigma\sqrt{\log(n)})$ around $\hat{\mu}_1$ to this subgroup and using the truncate-then-Laplace noise method of UVGAUSSTIMATE. The cost, due to the $g = O\left(\left[\log\left(\frac{\sigma_{\max}}{\sigma_{\min}}\right)+1\right]\sqrt{\log(n)}\right)$ subgroups required, is the $1/\sqrt{g}$ reduction in accuracy shown in Theorem 4.2.

## 5 Lower Bound

We now show that all of our upper bounds are tight up to logarithmic factors. Our argument has three steps: we first reduce our estimation problem to a testing problem, then reduce this testing problem to a purely locally private testing problem, and finally prove a lower bound for this purely locally private testing problem. Taken together, these results show that estimation is hard for sequentially interactive $(\varepsilon, \delta)$-locally private protocols. An extension to fully interactive protocols using recent subsequent work by Joseph et al. [16] appears in the full version of this paper [15].

**Theorem 5.1.** *Let* $\delta < \min\left(\frac{\epsilon\beta}{60n\ln(5n/2\beta)}, \frac{\beta}{16n\ln(n/\beta)e^{7\varepsilon}}\right)$ *and* $\varepsilon > 0$. *There exists absolute constant* $c$ *such that if* $\mathcal{A}$ *is an* $(\varepsilon, \delta)$-*locally private* $(\alpha, \beta)$-*estimator for* $\mathsf{Estimate}\,(n, M, \sigma)$ *where* $M = \sigma/[4(e^\varepsilon - 1)\sqrt{2nc}]$ *and* $\beta < 1/16$, *then* $\alpha \geq M/2 = \Omega\left(\frac{\sigma}{\varepsilon}\sqrt{\frac{1}{n}}\right)$.

## Footnotes

[4]As "adaptive" and "nonadaptive" are implicit in "two-round" and "one-round", we often omit these terms.

[5]The notion of a central coordinating analyst is only a useful simplification. As the analyst has no special powers or privileges, any user, or the protocol itself, can be viewed as playing the same role.

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
