[Supplementary Material]

# 1 Proofs from Section 3.1

We start with pseudocode for RR1.

---

**Algorithm 1** RR1

---

**Input:** $\varepsilon, i, j$
1: $y_i \leftarrow \lfloor x_i/2^j \rfloor \bmod 4$
2: **if** $p \sim_U [0,1] \leq \frac{e^\varepsilon}{e^\varepsilon+3}$ **then**
3:     User $i$ publishes $\tilde{y}_i \leftarrow y_i$
4: **else**
5:     User $i$ publishes $\tilde{y}_i \sim_u (\{0,1,2,3\}\backslash\{y_i\})$
6: **end if**
**Output:** Private user estimate $\tilde{y}_i$ of $\mu(j)$

---

Next, we prove the privacy guarantee for KVGAUSSTIMATE.

**Theorem 1.1.** KVGAUSSTIMATE *satisfies* $(\varepsilon, 0)$*-local differential privacy for* $x_1, \ldots, x_n$.

*Proof.* As KVGAUSSTIMATE is sequentially interactive, each user only produces one output. It therefore suffices to show that each randomized response routine used in KVGAUSSTIMATE is $(\varepsilon, 0)$-locally private. In RR1, for any possible inputs $x, x'$ and output $y$ we have

$$\frac{\mathbb{P}\left[\text{RR}1(x) = y\right]}{\mathbb{P}\left[\text{RR}1(x') = y\right]} \leq \frac{e^\varepsilon/(e^\varepsilon+3)}{1/(e^\varepsilon+3)} \leq e^\varepsilon$$

so RR1 is $(\varepsilon, 0)$-locally private. KVRR2 is $(\varepsilon, 0)$-locally private by similar logic. $\square$

We now prove the accuracy guarantee for KVGAUSSTIMATE. First, recall that $\hat{H}_1$ is the aggregation (via KVAGG1) of user responses (via RR1). Let $H_1$ be the "true" histogram, $H_1^j(a) = |\{y_i \mid i \in U_1^j, y_i = a\}|$ for all $a \in \{0,1,2,3\}$ and $j \in \mathcal{L}$. Since the analyst only has access to $\hat{H}_1$, we need to show that $\hat{H}_1$ and $H_1$ are similar.

**Lemma 1.2.** *With probability at least* $1 - \beta$, *for all* $j \in \mathcal{L}$,

$$\|\hat{H}_1^j - H_1^j\|_\infty \leq \left(\tfrac{\varepsilon+4}{\varepsilon\sqrt{2}}\right) \cdot \sqrt{k\ln(8L/\beta)}.$$

*Proof.* Choose $a \in \{0,1,2,3\}$ and $j \in \mathcal{L}$. $\mathbb{E}\left[C^j(a)\right] = \frac{H_1^j(a)e^\varepsilon}{e^\varepsilon+3} + \frac{k-H_1^j(a)}{e^\varepsilon+3} = \frac{H_1^j(a)(e^\varepsilon-1)+k}{e^\varepsilon+3}$, so by a pair of Chernoff bounds on the $k$ users in $U_1^j$, with probability at least $1 - \beta/4L$,

$$|C^j(a) - \tfrac{H_1^j(a)(e^\varepsilon-1)+k}{e^\varepsilon+3}| \leq \sqrt{k\ln(8L/\beta)/2}.$$

Then since $\hat{H}_1^j(a) = \frac{e^\varepsilon+3}{e^\varepsilon-1} \cdot \left(C^j(a) - \frac{k}{e^\varepsilon+3}\right)$, this implies

$$|\hat{H}_1^j(a) - H_1^j(a)| \leq \frac{e^\varepsilon+3}{e^\varepsilon-1} \cdot \sqrt{k\ln(8L/\beta)/2} < \left(\tfrac{\varepsilon+4}{\varepsilon\sqrt{2}}\right) \cdot \sqrt{k\ln(8L/\beta)}$$

where the last step uses $\frac{e^\varepsilon+3}{e^\varepsilon-1} < \frac{\varepsilon+4}{\varepsilon}$. Union bounding over $a \in \{0,1,2,3\}$ and all $L$ groups $U_1^j$ completes the proof. $\square$

Next, we show how the analyst uses $\hat{H}_1$ to estimate $\mu$ through ESTMEAN. Intuitively, in subgroup $U_1^j$ when user responses concentrate in a single bin mod 4, this suggests that $\mu$ lies in the corresponding bin. In the other direction, when user responses do not concentrate in a single bin, users with points near $\mu$ must spread out over multiple bins, suggesting that $\mu$ lies near the boundary between bins. We formalize this intuition in ESTMEAN and Lemma 1.3.

**Lemma 1.3.** *Conditioned on the success of the preceding lemmas, with probability at least* $1 - \beta$, $|\hat{\mu}_1 - \mu| \leq 2\sigma$.

*Proof.* Recall the definitions of $\psi$, $M_1(j)$, and $M_2(j)$ from the pseudocode for EST-MEAN: $\psi = \left(\frac{\varepsilon+4}{\varepsilon\sqrt{2}}\right) \cdot \sqrt{k\ln(8L/\beta)}$, $M_1(j) = \arg\max_{a\in\{0,1,2,3\}} \hat{H}_1^j(a)$, and $M_2(j) = \arg\max_{a\in\{0,1,2,3\}-\{M_1(j)\}} \hat{H}_1^j(a)$. We start by proving two useful claims.

Claim 1: With probability at least $1-\beta/5$, for all $j \in \mathcal{L}$ where $2^j > \sigma$, if $j' = L_{\max}, L_{\max}-1, \ldots, j+1$ all have $\hat{H}_1^{j'}(M_1(j)) \geq 0.52k + \psi$, then $\mu \in I_j$.

To see why, suppose $2^j > \sigma$ and let $x \sim N(\mu, \sigma^2)$. Recall the Gaussian CDF $F(x) = \frac{1}{2}\left[1 + \text{erf}\left(\frac{x-\mu}{\sigma\sqrt{2}}\right)\right]$. Then for any $a \not\equiv \lfloor \mu/2^j \rfloor \mod 4$

$$\mathbb{P}\left[\lfloor x/2^j \rfloor \equiv a \mod 4\right] \leq \mathbb{P}\left[x \notin [\mu, \mu + 3 \cdot 2^j]\right] < \mathbb{P}\left[x \notin [\mu, \mu + 3\sigma]\right] < 0.51$$

where the second inequality uses $2^j > \sigma$. Thus by a binomial Chernoff bound, the assumption $k > 5000\ln(5L/\beta)$, and Lemma 1.2, with probability $\geq 1 - \beta/5L$, $\hat{H}_1^j(a) < 0.52k + \psi$. Therefore if for some $a$ we have $\hat{H}_1^j(a) \geq 0.52k + \psi$, $a \equiv \lfloor \mu/2^j \rfloor \mod 4$. Moreover, if $\mu \in I_j$ then letting $c$ be the (unique) integer such that $c \equiv M_1(j) \mod 4$ and $c2^j \in I_j$ (since $I_j$ has endpoints $c_1 2^j$ and $(c_1 + 2)2^j$ for integer $c_1$) we get $\mu \in [c2^j, (c+1)2^j] = I_j$. As $\mu \in I_{L_{\max}}$ by our assumed lower bound on $n$, the claim follows by induction.

Claim 2: Let $j$ be the maximum $j \in \mathcal{L}$ with $\hat{H}_1^j(M_1(j)) < 0.52k + \psi$, and let $c^*$ be the maximum integer such that $c^*2^j \in I_j$ and $c^* \equiv M_1(j)$ or $M_2(j) \mod 4$. If $2^j > \sigma$, then with probability at least $1 - 4\beta/5$, $|c^*2^j - \mu| \leq 2\sigma$.

To see why, first note that by Claim 1, $\mu \in I_j$. Let $[c2^j, (c+1)2^j)$ be the subinterval of $I_j$ containing $\mu$ for integer $c$. Then as $2^j > \sigma$, for $x \sim N(\mu, \sigma^2)$, by another application of the Gaussian CDF,

$$\mathbb{P}\left[x \in [c2^j, (c+1)2^j)\right] > \mathbb{P}\left[x \in [\mu, \mu + \sigma]\right] \geq 0.34.$$

Thus by the same method as above, using the assumption $k > 5000\ln(5/\beta)$, with probability at least $1 - \beta/5$, $\hat{H}_1^j(c \mod 4) \geq 0.33k - \psi$. By similar logic, since

$$\mathbb{P}\left[\lfloor x/2^j \rfloor \equiv c + 2 \mod 4\right] < \max_{\lambda \in [0,2^j]} \mathbb{P}\left[x \notin [\mu - 2^j - \lambda, \mu + 2 \cdot 2^j - \lambda]\right] < \mathbb{P}\left[x \notin [\mu - \sigma, \mu + 2\sigma]\right] \leq 0.19$$

with probability at least $1 - \beta/5$, $\hat{H}_1^j(c + 2 \mod 4) \leq 0.2k + \psi$. Next, consider $\hat{H}_1^j(c - 1 \mod 4)$. If $\mu \geq (c + 0.75)2^j$, then

$$\mathbb{P}\left[x \in [(c-1)2^j, c2^j)\right] \leq \mathbb{P}\left[x \notin [\mu - 3\sigma/4, \mu + 9\sigma/4]\right] \leq 0.24$$

so with probability at least $1 - \beta/5$

$$\hat{H}_1^j(c - 1 \mod 4) \leq 0.25k + \psi < 0.33k - \psi \leq \hat{H}_1^j(c \mod 4)$$

where the middle inequality uses $k > 625\left(\frac{\varepsilon+4}{\varepsilon\sqrt{2}}\right)^2 \ln(4L/\beta)$. Thus $c \equiv M_1(j)$ or $M_2(j) \mod 4$; the $\mu \leq (c + 0.25)2^j)$ case is symmetric. If instead $\mu \in ((c + 0.25)2^j, (c + 0.75)2^j)$ then by similar logic with probability at least $1 - \beta/5$

$$\hat{H}_1^j(c \mod 4) \geq 0.36k - \psi.$$

so by $\psi < 0.08k$ (implied by $k > 40\left(\frac{\varepsilon+4}{\varepsilon\sqrt{2}}\right)^2 \ln(8L/\beta)$) $c \equiv M_1(j)$ or $M_2(j) \mod 4$. It follows that with probability at least $1 - 3\beta/5$ in all cases $c \equiv M_1(j)$ or $M_2(j) \mod 4$. Moreover, by a similar application of the Gaussian CDF, one of $c - 1 \mod 4$ and $c + 1 \mod 4$ lies in $\{M_1(j), M_2(j)\}$ as well.

Recalling that $c^*$ is the maximum integer such that $c^*2^j \in I_j$ and $c^* \equiv M_1(j)$ or $M_2(j) \mod 4$, $c^* - 1 \mod 4 \in \{M_1(j), M_2(j)\}$ as well. Assume $|c^*2^j - \mu| > 2\sigma$. By above, $\mu \in [c^*2^j, (c^* + 1)2^j)$ or $[(c^* - 1)2^j, (c^*2^j))$. In the first case,

$$\mathbb{P}\left[\lfloor x/2^j \rfloor \equiv c^* - 1 \mod 4\right] \leq \mathbb{P}\left[x \notin [\mu - 2\sigma, \mu + 2\sigma]\right] \leq 0.05$$

59　so with probability at least $1 - \beta/5$, $\hat{H}_1^j(c^* - 1) \leq 0.06k + \psi$, a contradiction of $c^* - 1 \bmod 4 \in$
60　$\{M_1(j), M_2(j)\}$. In the second case,

$$\mathbb{P}\left[\lfloor x/2^j \rfloor \equiv c^* \bmod 4\right] \leq \mathbb{P}\left[x \notin [\mu - 2\sigma, \mu + 2\sigma]\right] \leq 0.05$$

61　and with probability at least $1 - \beta/5$, $\hat{H}_1^j(c^*) \leq 0.06k + \psi$, contradicting $c^* \bmod 4 \in \{M_1(j), M_2(j)\}$.
62　Thus $|c^* 2^j - \mu| \leq 2\sigma$.

63　We put these facts together in ESTMEAN as follows: let $j_1$ be the maximum element of $\mathcal{L}$ such that
64　$\hat{H}_1^j(M_1(j)) < 0.52k - \psi$. If $2^{j_1} > \sigma$, then by Fact 2 setting $\hat{\mu}_1 = c^* 2^j$ implies $|\hat{\mu}_1 - \mu| \leq 2\sigma$. If
65　instead $2^{j_1} \leq \sigma$, then any setting of $\hat{\mu}_1 \in I_j$ (including $\hat{\mu}_1 = c^* 2^j$) guarantees $|\hat{\mu}_1 - \mu| \leq 2^{j_1+1} \leq 2\sigma$.
66　Thus in all cases, with probability at least $1 - \beta$, $|\hat{\mu}_1 - \mu| \leq 2\sigma$.　　　　　$\square$

67　The results above give the analyst an (initial) estimate $\hat{\mu}_1$ such that $|\hat{\mu}_1 - \mu| \leq 2\sigma$. This concludes our
68　analysis of round one of KVGAUSSTIMATE. Now, the analyst passes this estimate $\hat{\mu}_1$ to users $i \in U_2$,
69　and each user uses $\hat{\mu}_1$ to center their value $x_i$ and randomized respond on the resulting $(x_i - \hat{\mu}_1)/\sigma$ in
70　KVRR2. The analyst then aggregates these results using KVAGG2. We now prove that this centering
71　process results in a more accurate final estimate $\hat{\mu}_2$ of $\mu$.

72　**Lemma 1.4.** *Conditioned on the success of the previous lemmas, with probability at least $1 - \beta$*
73　KVGAUSSTIMATE *outputs $\hat{\mu}_2$ such that*

$$|\hat{\mu}_2 - \mu| = O\left(\frac{\sigma}{\varepsilon}\sqrt{\frac{\log(1/\beta)}{n}}\right).$$

74　*Proof.* The proof is broadly similar to that of Theorem B.1 in Braverman et al. [1], with some
75　modifications for privacy. First, by Lemma 1.3 $\mu - \hat{\mu}_1 \in [-2\sigma, 2\sigma]$. Letting $\bar{\mu} = (\mu - \hat{\mu}_1)/\sigma$ we get
76　that $x_i' \sim N(\bar{\mu}, 1)$. Next, since $\mathbb{E}[y_i] = 2\mathbb{P}[x_i' \geq 0] - 1$, and in general

$$\Phi_{\mu,\sigma^2}(x) = \frac{1}{2}\left(1 + \mathrm{erf}\left(\frac{x - \mu}{\sigma\sqrt{2}}\right)\right)$$

77　where $\Phi_{\mu,\sigma^2}$ is the CDF of $N(\mu, \sigma^2)$, by $\Phi_{\bar{\mu},1}(0) = \mathbb{P}[x_i' \geq 0]$ we get $\mathbb{E}[y_i] = \mathrm{erf}(\bar{\mu}/\sqrt{2})$ (note
78　that we are analyzing the unprivatized values $y_i$ to start; later, we will use this analysis to prove the
79　analogous result for the privatized values $\tilde{y}_i$).

80　A Chernoff bound on $[-1, 1]$-bounded random variables then shows that, with probability at least
81　$1 - \beta/2$, for $y = \frac{2}{n}\sum_{i \in U_2} y_i$ we have

$$|y - \mathrm{erf}(\bar{\mu}/\sqrt{2})| \leq 2\sqrt{\ln(4/\beta)/n}$$

82　and by $\mathbb{E}[y] = \mathrm{erf}(\bar{\mu}/\sqrt{2})$ we get $|y - \mathbb{E}[y]| \leq 2\sqrt{\ln(4/\beta)/n}$ as well.

83　Since $\mu - \hat{\mu}_1 \in [-2\sigma, 2\sigma]$, $|\mathrm{erf}(\bar{\mu}/\sqrt{2})| \leq \mathrm{erf}(\sqrt{2})$. Thus $|\mathbb{E}[y]| \leq \mathrm{erf}(\sqrt{2})$, so by $|y - \mathbb{E}[y]| \leq$
84　$2\sqrt{\ln(4/\beta)/n}$ we get

$$|y| \leq \mathrm{erf}(\sqrt{2}) + 2\sqrt{\ln(4/\beta)/n}.$$

85　Using $n > 20000\ln(4/\beta)$ we get $2\sqrt{\ln(4/\beta)/n} < 0.01$ and $\mathrm{erf}(\sqrt{2}) < 0.96$, so $|y| \leq 0.97$ and thus
86　$|y| < \mathrm{erf}(1.6)$. Let $M$ be an upper bound on the Lipschitz constant for $\mathrm{erf}^{-1}$ in $[-0.97, 0.97]$,

$$M = \max_{x \in [-0.97, 0.97]} \frac{d\,\mathrm{erf}^{-1}(x)}{dx}$$

$$= \max_{x \in [-0.97, 0.97]} \frac{\sqrt{\pi}}{2}\exp([\mathrm{erf}^{-1}(x)]^2)$$

$$\leq \frac{\sqrt{\pi}}{2}\exp([\mathrm{erf}^{-1}(0.97)]^2) < 10.$$

87　Then for any $x, y \in [-0.97, 0.97]$ we have $|\mathrm{erf}^{-1}(x) - \mathrm{erf}^{-1}(y)| \leq M|x - y|$, so setting $T =$
88　$\sqrt{2}\,\mathrm{erf}^{-1}(y)$,

$$|T - \bar{\mu}| = |\sqrt{2}(\mathrm{erf}^{-1}(y) - \mathrm{erf}^{-1}(\mathbb{E}[y]))| \leq 10\sqrt{2}|y - \mathbb{E}[y]|$$

$$\leq 20\sqrt{2\ln(4/\beta)/n}$$

89     using the bound on $|y - \mathbb{E}[y]|$ from above.

90     It remains to analyze the privatized values $\{\tilde{y}_i\}$ and bound $|T - \hat{T}|$, recalling that we set

$$\hat{T} = \sqrt{2} \cdot \operatorname{erf}^{-1}\left(\frac{2(-\hat{H}_2(-1) + \hat{H}_2(1))}{n}\right)$$

91     in KVAGG1. By a Chernoff bound analogous to that of Lemma 1.2, with probability at least $1 - \beta/2$

$$|T - \hat{T}| \leq \sqrt{2}\left|\operatorname{erf}^{-1}(|y|) - \operatorname{erf}^{-1}\left(|y| + \left[\frac{\varepsilon + 2}{\varepsilon}\right]\sqrt{\frac{2\ln(4/\beta)}{n}}\right)\right|.$$

92     Using $n > 20000\left(\frac{\varepsilon+2}{\varepsilon}\right)^2 \ln(4/\beta)$ (which implies $\left[\frac{\varepsilon+2}{\varepsilon}\right]\sqrt{\frac{2\ln(4/\beta)}{n}} \leq 0.01$) and the same derivative
93     trick as above on $[-0.98, 0.98]$, we get

$$|T - \hat{T}| \leq 14\left[\frac{\varepsilon + 2}{\varepsilon}\right]\sqrt{\frac{2\ln(4/\beta)}{n}}.$$

94     Therefore by the triangle inequality

$$|\hat{T} - \bar{\mu}| \leq \left(20 + 14\left[\frac{\varepsilon + 2}{\varepsilon}\right]\right)\sqrt{\frac{2\ln(4/\beta)}{n}}$$

95     and by $\sigma\bar{\mu} = \mu - \hat{\mu}_1$ we get

$$|\sigma\hat{T} - \sigma\bar{\mu}| = |(\sigma\hat{T} + \hat{\mu}_1) - \mu| \leq \sigma\left(20 + 14\left[\frac{\varepsilon + 2}{\varepsilon}\right]\right)\sqrt{\frac{2\ln(4/\beta)}{n}}.$$

96     Thus by taking $\hat{\mu}_2 = \sigma\hat{T} + \hat{\mu}_1$, we get

$$|\hat{\mu}_2 - \mu| = O\left(\frac{\sigma}{\varepsilon}\sqrt{\frac{\log(1/\beta)}{n}}\right).$$

97     $\square$

## 2   Proofs from Section 3.2

99     We start with full pseudocode for 1ROUNDKVGAUSSTIMATE.

---

**Algorithm 2** 1ROUNDKVGAUSSTIMATE

---

**Input:** $\varepsilon, k_1, k_2, \mathcal{L}, n, R, S, \sigma, U_1, U_2$
  1: **for** $j \in \mathcal{L}$ **do**
  2:      **for** user $i \in U_1^j$ **do**
  3:        User $i$ outputs $\tilde{y}_i \leftarrow \text{RR1}(\varepsilon, i, j)$
  4:      **end for**
  5: **end for**
  6: **for** $j \in R$ **do**
  7:      **for** user $i \in U_2^j$ **do**
  8:        User $i$ outputs $\tilde{y}_i \leftarrow \text{1ROUNDKVRR2}(\varepsilon, i, S(j))$
  9:      **end for**
10: **end for**                              ▷ End of round 1
11: Analyst computes $\hat{H}_1 \leftarrow \text{KVAGG1}(\varepsilon, k_1, \mathcal{L}, U_1)$
12: Analyst computes $\hat{\mu}_1 \leftarrow \text{ESTMEAN}(\beta, \varepsilon, \hat{H}_1, k_1, \mathcal{L}, )$
13: Analyst computes $j^* \leftarrow \arg\min_{j \in R} \min_{s \in S(j)} |s - \hat{\mu}_1|$
14: Analyst computes $\hat{H}_2 \leftarrow \text{KVAGG2}(\varepsilon, k_2, U_2^{j^*})$
15: Analyst computes $\hat{T} \leftarrow \sqrt{2} \cdot \operatorname{erf}^{-1}\left(\frac{-\hat{H}_2(-1) + \hat{H}_2(1)}{k_2}\right)$
16: Analyst outputs $\hat{\mu}_2 \leftarrow \sigma\hat{T} + \arg\min_{s \in S(j^*)} |s - \hat{\mu}_1|$
**Output:** Analyst estimate $\hat{\mu}_2$ of $\mu$

---

100 1ROUNDKVGAUSSTIMATE's privacy guarantee follows from the same analysis of randomized
101 response as in KVGAUSSTIMATE, so we state the guarantee but omit its proof.

102 **Theorem 2.1.** 1ROUNDKVGAUSSTIMATE *satisfies* $(\varepsilon, 0)$-*local differentially privacy for* $x_1, \ldots, x_n$.

103 We define $k$ (here denoted $k_1$), $\mathcal{L}, U_1$, and $U_2$ as in KVGAUSSTIMATE. As 1ROUNDKVGAUSSTI-
104 MATE's treatment of users in $U_1$ is identical to that of KVGAUSSTIMATE, we skip its analysis, instead
105 recalling its final guarantee:

106 **Lemma 2.2.** *With probability at least* $1 - \beta$, $|\hat{\mu}_1 - \mu| \leq 2\sigma$.

107 This brings us to $U_2$, and we define new parameters as follows. For neatness, let $\rho = \lceil 2\sqrt{\ln(4n)} \rceil \geq$
108 $\lceil \sqrt{2\ln(2\sqrt{n})} + 2.1 \rceil$ for $n \geq 32$. We set $R = \{0.2\sigma, 0.4\sigma, \ldots, \rho\sigma\}$ and split $U_2$ into $|R| = 5\rho$ groups
109 indexed by $j \in R$, each of size $k_2 \geq \lfloor n/2|R| \rfloor \geq \lfloor \frac{n}{20\sqrt{\ln(4n)}} \rfloor = \Omega(n/\sqrt{\log(n)})$, where the last
110 inequality uses $n \geq 25$. Finally, for each $j \in R$ we define $S(j) = \{j + b\rho\sigma \mid b \in \mathbb{Z}\}$.

111 With this setup, for each $j \in R$ each user $i \in U_2^j$ uses 1ROUNDKVRR2 to execute a group-specific
112 version of KVRR2: rather than centering by $\hat{\mu}_1$ as in KVRR2, user $i$ now centers by the nearest
113 point in $S(j)$ (breaking ties arbitrarily).

---

**Algorithm 3** 1ROUNDKVRR2

**Input:** $\varepsilon, i, S(j)$
1: User $i$ computes $z_i \leftarrow \arg\min_{z_i \in S(j)} |z_i - x_i|$
2: User $i$ computes $y_i \leftarrow \operatorname{sgn}((x_i - z_i)/\sigma)$
3: User $i$ computes $c \sim_U [0, 1]$
4: **if** $c \leq \frac{e^\varepsilon}{e^\varepsilon + 1}$ **then**
5:     User $i$ publishes $\tilde{y}_i \leftarrow y_i$
6: **else**
7:     User $i$ publishes $\tilde{y}_i \leftarrow -y_i$
8: **end if**
**Output:** Private centered user estimate $\tilde{y}_i$

---

114 To analyze 1ROUNDKVRR2, we first prove that users in each group draw points concentrated around
115 $\mu$.

116 **Lemma 2.3.** *With probability at least* $1 - \beta$, *for all* $j \in R$, *group* $U_2^j$ *contains* $\leq 2\sqrt{k_2}$ *users* $i$ *such*
117 *that* $|x_i - \mu| > \sigma\sqrt{\ln(4n)}$.

118 *Proof.* First, by a Gaussian tail bound, for each user $i$, $\mathbb{P}\left[|x_i - \mu| \geq \sigma\sqrt{\ln(4n)}\right] \leq 1/\sqrt{n}$. Let $U_C^j$
119 denote the users in group $U_2^j$ such that $|x_i - \mu| > \sigma\sqrt{\ln(4n)}$. Then by a binomial Chernoff bound

$$\mathbb{P}\left[|U^c| > \frac{k_2}{\sqrt{n}} + \sqrt{\frac{3k_2\ln(|R|/\beta)}{\sqrt{n}}}\right] \leq \beta/|R|$$

120 so using $n \geq 9\ln(|R|/\beta)^2$ and union bounding over $|R| = \Omega(\sqrt{\log(n)})$ groups, the claim follows. $\square$

121 In particular, this implies that for $j^* = \arg\min_{j^* \in R} \min_{s \in S(j^*)} |s - \hat{\mu}_1|$ (i.e., the group with element
122 of $S(j^*)$ closest to $\hat{\mu}_1$), most users draw points in $[\mu - \sigma\sqrt{\ln(4n)}, \mu + \sigma\sqrt{\ln(4n)}]$. Let $s^* =$
123 $\min_{s \in S(j^*)} |s - \hat{\mu}_1|$. Our final accuracy result will rely on two facts. First, most users in $U_2^{j^*}$ center
124 using $s^*$. Second, the randomized responses of users who center with $s^*$ are "almost as good" as if
125 they were centered by $\mu$.

126 **Lemma 2.4.** *Conditioned on the success of the previous lemmas, with probability at least* $1 - \beta$,
127 1ROUNDKVGAUSSTIMATE *outputs* $\hat{\mu}_2$ *such that*

$$|\hat{\mu}_2 - \mu| = O\left(\frac{\sigma}{\varepsilon}\sqrt{\frac{\log(1/\beta)\sqrt{\log(n)}}{n}}\right).$$

*Proof.* Because adjacent points in $R$ are $0.2\sigma$ apart, $|s^* - \hat{\mu}_1| \leq 0.1\sigma$. Lemma 2.2 and the triangle inequality then imply that $|s^* - \mu| \leq 2.1\sigma$. This enables us to mimic the proof of Lemma 1.4, replacing $\mu - \hat{\mu}_1 \in [-2\sigma, 2\sigma]$ with $\mu - s^* \in [-2.1\sigma, 2.1\sigma]$.

We can decompose users in $U_2^{j^*}$ into those with points within $\sigma\rho$ of $s^*$ and those with more distant points. Denote the first set of users by $V$ and the second set by $V^c$, and recall that the Gaussian CDF is

$$\Phi_{\mu,\sigma^2}(x) = \frac{1}{2}\left(1 + \text{erf}\left(\frac{x - \mu}{\sigma\sqrt{2}}\right)\right).$$

Then, letting $\mathbb{1}$ denote the indicator function,

$$
\begin{aligned}
\mathbb{E}\left[y_i \cdot \mathbb{1}(i \in V)\right] &= \mathbb{P}\left[y_i = 1, i \in V\right] - \mathbb{P}\left[y_i = -1, i \in V\right] \\
&= \Phi_{\mu,\sigma^2}(s^* + \sigma\rho) + \Phi_{\mu,\sigma^2}(s^* - \sigma\rho) - 2\Phi_{\mu,\sigma^2}(s^*) \\
&= \frac{1}{2}\left[\text{erf}\left(\frac{s^* + \sigma\rho - \mu}{\sigma\sqrt{2}}\right) + \text{erf}\left(\frac{s^* - \sigma\rho - \mu}{\sigma\sqrt{2}}\right)\right] - \text{erf}\left(\frac{s^* - \mu}{\sigma\sqrt{2}}\right) \\
&= \frac{1}{2}\left[\text{erf}\left(\frac{\sigma\rho + s^* - \mu}{\sigma\sqrt{2}}\right) - \text{erf}\left(\frac{\sigma\rho - (s^* - \mu)}{\sigma\sqrt{2}}\right)\right] - \text{erf}\left(\frac{s^* - \mu}{\sigma\sqrt{2}}\right).
\end{aligned}
$$

where the last step uses the fact that erf is an odd function. Since $\text{erf}(x) = \frac{2}{\sqrt{\pi}}\int_0^x e^{-t^2}dt$ and $|s^* - \mu| \leq 2.1\sigma$,

$$
\begin{aligned}
\frac{1}{2}\left[\text{erf}\left(\frac{\sigma\rho + s^* - \mu}{\sigma\sqrt{2}}\right) - \text{erf}\left(\frac{\sigma\rho - (s^* - \mu)}{\sigma\sqrt{2}}\right)\right] &\leq \frac{1}{\sqrt{\pi}}\int_{(\sigma\rho - 2.1\sigma)/\sigma\sqrt{2}}^{(\sigma\rho + 2.1\sigma)/\sigma\sqrt{2}} e^{-t^2}dt \\
&< 3e^{-[(\rho - 2.1)/\sqrt{2}]^2} \\
&\leq 3e^{-\ln(4n)/2}
\end{aligned}
$$

where the second inequality relies on $e^{-x}$ being monotone decreasing and the last step uses $n > 20$, which implies $\rho - 2.1 \geq \sqrt{\ln(4n)}$. Then using $n \geq 3k_2$ we get $3e^{-\ln(4n)/2} \leq \frac{1}{\sqrt{k_2}}$, so

$$\left|\mathbb{E}\left[y_i \cdot \mathbb{1}(i \in V)\right] - \text{erf}\left(\frac{\mu - s^*}{\sigma\sqrt{2}}\right)\right| \leq \frac{1}{\sqrt{k_2}}. \tag{1}$$

Next, as $|s^* - \mu| \leq 2.1\sigma$, users having points within $\sigma\sqrt{2\ln(2\sqrt{n})}$ of $\mu$ have points within $\sigma\rho$ of $s^*$. The Gaussian tail bound from Lemma 2.3 then implies $\mathbb{P}\left[x \in V^c\right] \leq 1/\sqrt{n}$. $\mathbb{E}\left[y_i\right] = \mathbb{E}\left[y_i \cdot \mathbb{1}(i \in V)\right] + \mathbb{E}\left[y_i \cdot \mathbb{1}(i \in V^c)\right]$, and by the above bound on $\mathbb{P}\left[x \in V^c\right]$ and $|y_i| \leq 1$ we get $|\mathbb{E}\left[y_i \cdot \mathbb{1}(i \in V^c)\right]| \leq 1/\sqrt{n}$. Thus

$$\left|\mathbb{E}\left[y_i \cdot \mathbb{1}(i \in V)\right] - \mathbb{E}\left[y_i\right]\right| \leq \frac{1}{\sqrt{n}} < \frac{1}{\sqrt{k_2}}. \tag{2}$$

A Chernoff bound on $\{-1, 1\}$-valued random variables then tells us that, for $y = \frac{1}{k_2}\sum_{i \in U_2^{j^*}} y_i$, with probability at least $1 - \beta/2$ we have

$$|y - \mathbb{E}\left[y_i\right]| \leq \sqrt{\frac{2\ln(4/\beta)}{k_2}}. \tag{3}$$

Combining the three numbered equations above with the triangle inequality yields

$$\left|y - \text{erf}\left(\frac{\mu - s^*}{\sigma\sqrt{2}}\right)\right| < \frac{2 + \sqrt{2\ln(4/\beta)}}{\sqrt{k_2}}.$$

Setting $\bar{\mu} = (\mu - s^*)/\sigma$ and using $k_2 \geq (100[2 + \sqrt{2\ln(4/\beta)}])^2$, this rearranges into $|y| \leq \text{erf}(\bar{\mu}/\sqrt{2}) + 0.01$. Since $\bar{\mu} \in [-2.1, 2.1]$, we get

$$|y| < \text{erf}(2.1/\sqrt{2}) + 0.01 < 0.98 < \text{erf}(1.7).$$

148  Let $M$ be an upper bound on the Lipschitz constant for $\text{erf}^{-1}$ in $[-0.98, 0.98]$,

$$M = \max_{x \in [-0.98, 0.98]} \frac{d\text{erf}^{-1}(x)}{dx}$$

$$= \max_{x \in [-0.98, 0.98]} \frac{\sqrt{\pi}}{2} \exp([\text{erf}^{-1}(x)]^2)$$

$$\leq \frac{\sqrt{\pi}}{2} \exp([\text{erf}^{-1}(0.98)]^2) < 14.$$

149  Then for any $x, y \in [-0.98, 0.98]$ we have $|\text{erf}^{-1}(x) - \text{erf}^{-1}(y)| \leq M|x - y|$, so for $T = \sqrt{2}\text{erf}^{-1}(y)$,

$$|T - \bar{\mu}| = |\sqrt{2}(\text{erf}^{-1}(y) - \text{erf}^{-1}(\text{erf}(\bar{\mu}/\sqrt{2})))| \leq 14\sqrt{2}|y - \text{erf}(\bar{\mu}/\sqrt{2})|$$

$$< 28\left(\frac{\sqrt{2} + \sqrt{\ln(4/\beta)}}{k_2}\right).$$

150  It remains to bound $|T - \hat{T}|$, where $T$ is the (unknown) aggregation of unprivatized $\{y_i\}$ while $\hat{T}$ is
151  the (known) aggregation of privatized $\{\tilde{y}_i\}$. By a Chernoff bound analogous to that of Lemma 1.2,
152  with probability at least $1 - \beta/2$

$$|T - \hat{T}| \leq \sqrt{2}\left|\text{erf}^{-1}(|y|) - \text{erf}^{-1}\left(|y| + \left[\frac{\varepsilon + 2}{\varepsilon}\right]\sqrt{\frac{2\ln(4/\beta)}{k_2}}\right)\right|.$$

153  Using $k_2 > 20000\left(\frac{\varepsilon+2}{\varepsilon}\right)^2 \ln(4/\beta)$ (which implies $\left[\frac{\varepsilon+2}{\varepsilon}\right]\sqrt{2\frac{\ln(4/\beta)}{k_2}} \leq 0.01$) and the same derivative
154  trick as above on $[-0.99, 0.99]$, we get

$$|T - \hat{T}| \leq 25\left[\frac{\varepsilon + 2}{\varepsilon}\right]\sqrt{\frac{2\ln(4/\beta)}{k_2}}.$$

155  Therefore by the triangle inequality

$$|\hat{T} - \bar{\mu}| \leq 28\left(\frac{\sqrt{2} + \sqrt{\ln(4/\beta)}}{k_2}\right) + 25\left[\frac{\varepsilon + 2}{\varepsilon}\right]\sqrt{\frac{2\ln(4/\beta)}{k_2}} = O\left(\frac{1}{\varepsilon}\sqrt{\frac{\log(1/\beta)}{k_2}}\right)$$

156  and by $\sigma\bar{\mu} = \mu - s^*$ we get

$$|\sigma\hat{T} - \sigma\bar{\mu}| = |(\sigma\hat{T} + s^*) - \mu| = O\left(\frac{\sigma}{\varepsilon}\sqrt{\frac{\log(1/\beta)}{k_2}}\right).$$

157  Thus by taking $\hat{\mu}_2 = \sigma\hat{T} + s^*$ and substituting in $k_2 = \Omega(n/\sqrt{\log(n)})$ we get

$$|\hat{\mu}_2 - \mu| = O\left(\frac{\sigma}{\varepsilon}\sqrt{\frac{\log(1/\beta)\sqrt{\log(n)}}{n}}\right).$$

158  $\qquad\qquad\qquad\qquad\qquad\qquad\qquad\qquad\qquad\qquad\qquad\qquad\qquad\qquad\qquad\qquad\quad\square$

## 159  3  Proofs from Section 4.1

160  We begin our analysis with a privacy guarantee.

161  **Theorem 3.1.** UVGAUSSTIMATE *satisfies $(\varepsilon, 0)$-local differentially privacy for $x_1, \ldots, x_n$.*

162  *Proof.* As we already proved that RR1 is private in Section 1, we are left with UVRR2. To prove that
163  UVRR2 is $(\varepsilon, 0)$-locally differentially private as well, we can use a standard Laplace noise privacy
164  guarantee (see e.g. Theorem 3.6 from Dwork and Roth [6]): given function $f$ with 1-sensitivity $\Delta f$,
165  computing $f(x) + \text{Lap}(\Delta f/\varepsilon)$ satisfies $(\varepsilon, 0)$-differential privacy. $\qquad\qquad\square$

First, for each $j \in \mathcal{L}_1$ and $i \in U_1^j$, user $i$ employs RR1 (see Section 1) to publish a privatized version of $\lfloor x/2^j \rfloor \bmod 4$. The analyst then constructs two slightly different aggregations of this data. To estimate $\sigma$, the analyst aggregates responses into $\hat{H}_1$ via AGG1, which is similar to KVAGG1 up to the choice of bins in the constructed histogram $\hat{H}_1$. Specifically, bins in $\hat{H}_1$ are grouped: points with value 0 count toward both bin $(0,1)$ and bin $(3,0)$, points with value 1 count toward both bin $(0,1)$ and $(1,2)$, and so on.

---

**Algorithm 4** AGG1

---

**Input:** $\varepsilon, k, \mathcal{L}, U$
 1: **for** $j \in \mathcal{L}$ **do**
 2:     **for** $a = 0, 1, 2, 3$ **do**
 3:         Analyst computes $C^j(a) \leftarrow |\{i \mid i \in U^j, \tilde{y}_i = a\}|$
 4:         Analyst computes $\hat{H}^j(a) \leftarrow \frac{e^\varepsilon + 3}{e^\varepsilon - 1} \cdot \left( C^j(a) - \frac{k}{e^\varepsilon + 3} \right)$
 5:     **end for**
 6:     **for** $a = 0, 1, 2, 3$ **do**
 7:         Analyst computes $\hat{H}_1^j(a) \leftarrow \hat{H}^j(a) + \hat{H}^j(a + 1 \bmod 4)$
 8:     **end for**
 9: **end for**
10: Analyst outputs $\hat{H}_1$
**Output:** Analyst aggregation $\hat{H}_1$ of private user estimates

---

At a high level, when $2^j \gg \sigma$, user responses in group $U_1^j$ appear concentrated in one element of $\{(0,1), (1,2), (2,3), (3,0)\}$. This is because user data comes from $N(\mu, \sigma^2)$, so if $2^j \gg \sigma$ then most user data falls within $3\sigma$ of $\mu$. Consequently, there exists $a \in \{0, 1, 2, 3\}$ such that most users draw points $x$ where $\lfloor x/2^j \rfloor \equiv a$ or $a + 1 \bmod 4$, and $\hat{H}_1^j$ is concentrated around bin $(a, a + 1 \bmod 4)$. Similarly, if $2^j \ll \sigma$ then user responses in group $U_1^j$ appear unconcentrated (for a more precise definition of "concentrated", see below).

Examining this transition from concentrated to unconcentrated responses in $\hat{H}_1^{L_{\max}}, \hat{H}_1^{L_{\max}-1}, \dots$ yields a rough estimate of when $2^j \gg \sigma$ versus when $2^j \ll \sigma$. By approximating when this change occurs, the analyst recovers an approximation of $\sigma$. This process is outlined in ESTVAR.

---

**Algorithm 5** ESTVAR

---

**Input:** $\beta, \varepsilon, \hat{H}_1, k_1, \mathcal{L}_1$
 1: Analyst computes $j \leftarrow$ minimum $j$ such that, for all $j' \geq j$, $\hat{H}_1^{j'}$ is concentrated
 2: **if** $j = \varnothing$ **then**
 3:     Analyst outputs $\hat{\sigma} \leftarrow 2^{L_{\max}}$
 4: **else**
 5:     Analyst outputs $\hat{\sigma} \leftarrow 2^j$
 6: **end if**
**Output:** Analyst estimate $\hat{\sigma}$ of $\sigma$

---

$\hat{H}_1$ is an estimate of the "true" histogram collection, $H^j(a) = |\{y_i \mid i \in U_1^j, y_i \in \{a, a + 1 \bmod 4\}\}|$ for all $j \in \mathcal{L}_1$. As in Lemma 1.2, we can show that $\hat{H}_1$ and $H_1$ are similar. As the proof is nearly identical, we omit it.

**Lemma 3.2.** *With probability at least $1 - \beta$, for all $j \in \mathcal{L}_1$,*

$$\|\hat{H}_1^j - H_1^j\|_\infty \leq \left(1 + \tfrac{4}{\varepsilon}\right)\sqrt{2k_1 \ln(8L_1/\beta)}.$$

Next, we show how the analyst uses $\hat{H}_1$ to estimate $\sigma$ in subroutine ESTVAR. Here, for neatness we shorthand

$$\tau = \sqrt{2k_1 \ln(2L_1/\beta)} + \left(1 + \tfrac{4}{\varepsilon}\right)\sqrt{2k_1 \ln(8L_1/\beta)}$$

and use the term *concentrated* to denote any histogram $\hat{H}_1^j$ such that $\min_{a \in \{0,1,2,3\}} \hat{H}_1^j(a) \leq 0.03k + \tau$ and the term *unconcentrated* to denote $\hat{H}_1^j$ where $\min_{a \in \{0,1,2,3\}} \hat{H}_1^j(a) \geq 0.04k - \tau$. As we

189 show below in Lemma 3.3, when $2^j \gg \sigma$, $\hat{H}_1^j$ is concentrated. Similarly, when $2^j \ll \sigma$, $\hat{H}_1^j$ is
190 unconcentrated. This transition enables the analyst to estimate $\sigma$.

191 **Lemma 3.3.** *Conditioned on the success of the preceding lemmas, with probability at least $1 - \beta$,*
192 ESTVAR *outputs $\hat{\sigma} \in [\sigma, 8\sigma]$.*

193 *Proof.* Choose $j \in \mathcal{L}_1$. Below, we reason about two (non-exhaustive) possibilities for $j$.

194 Case 1: $2^j \geq 4\sigma$. Then there exists $a \in \{0, 1, 2, 3\}$ and interval $I$ of length $2^{j+1} \geq 8\sigma$ containing
195 $[\mu - 2\sigma, \mu + 2\sigma]$ such that for all $x \in I$, $\lfloor x/2^j \rfloor \mod 4 \equiv a$ or $a + 1 \mod 4$. By similar application of
196 the Gaussian CDF as in Lemma 1.3, with probability at least $1 - \beta/2L_1$,

$$|\{x_i \mid x_i \in I, i \in U_1^j\}| \geq 0.97k_1 - \sqrt{2k_1 \ln(2L_1/\beta)}.$$

197 Thus by Lemma 3.2, $\hat{H}_1^j(a) \geq 0.97k_1 - \tau$. It follows that $\hat{H}_1^j(a + 2) \leq 0.03k_1 + \tau$. $2^j \geq 4\sigma$ thus
198 implies that histogram $\hat{H}_1^j$ is concentrated.

199 Case 2: $2^j \in [\sigma/2, \sigma]$. Choose $a \in \{0, 1, 2, 3\}$. Since $2^j \in [\sigma/2, \sigma]$ there exist at most three
200 subintervals $I_1, I_2, I_3 \subset [\mu - 2\sigma, \mu + 2\sigma]$ such that for all $x \in I = I_1 \cup I_2 \cup I_3$, $\lfloor x/2^j \rfloor \equiv a \mod 4$, and
201 $|I| \geq \sigma$. Let $x \sim N(\mu, \sigma^2)$. Then by a similar application of the Gaussian CDF as in Lemma 1.3,
202 since

$$\mathbb{P}[x \in I] \geq \mathbb{P}[x \in [\mu - 2\sigma, \mu - \sigma)] \geq 0.13$$

203 with probability $1 - \beta/8L_1$ at least $0.13k - \sqrt{2k_1 \ln(8L_1/\beta)}$ users from $U_1^j$ have points in $I$. Since
204 this held for arbitrary $a$, a union bound over all four possibilities of $a$ combined with Lemma 3.2
205 implies that, with probability at least $1 - \beta/2L_1$,

$$\min_{a \in \{0,1,2,3\}} \hat{H}_1^j(a) \geq 0.13k_1 - \tau.$$

206 $2^j \leq \sigma \leq 2^{j+1}$ thus implies that histogram $\hat{H}_1^j$ is uniform.

207 Union bounding both results over $j \in \mathcal{L}_1$, with $k_1 > 800\left(2 + \frac{4}{\varepsilon}\right)^2 \ln(8L_1/\beta)$, with probability $1 - \beta$
208 we have $0.13k - \tau > 0.03k + \tau$ for each $j \in \mathcal{L}_1$. Therefore for all $j \in \mathcal{L}_1$ if $2^j \geq 4\sigma$ then $\hat{H}_1^j$ will be
209 concentrated while if $2^{j+1} \geq \sigma \geq 2^j$ then $\hat{H}_1^j$ will be unconcentrated.

210 Let $j$ be the smallest $j \in \mathcal{L}_1$ such that $\hat{H}_1^j$ is concentrated and for all $j' > j$, $\hat{H}_1^{j'}$ is concentrated as
211 well. If no such $j$ exists, then we know $2^{L_{\max}} \geq \sigma \geq 2^{L_{\max}-2}$, take $\hat{\sigma} = 2^{L_{\max}}$, and we get $\hat{\sigma} \in [\sigma, 4\sigma]$.
212 If not, then by Case 1 above we know $2^j \leq 8\sigma$, and by Case 2 we know $2^j \geq \sigma$. Thus taking $\hat{\sigma} = 2^j$,
213 we get $\hat{\sigma} \in [\sigma, 8\sigma]$. □

214 Next, the analyst uses randomized responses from $U_1$ to compute an initial estimate $\hat{\mu}_1$ of $\mu$. As the
215 process ESTMEAN is identical to that used in KVGAUSSTIMATE up to a different subgroup range
216 $\mathcal{L}_1$, we skip its description and only recall its guarantee:

217 **Lemma 3.4.** *Conditioned on the success of the preceding lemmas, with probability at least $1 - \beta$,*
218 $|\hat{\mu}_1 - \mu| \leq 2\sigma$.

219 From the results above, the analyst obtains an estimate $\hat{\sigma}$ such that $\hat{\sigma} \in [\sigma, 8\sigma]$ and an estimate $\hat{\mu}_1$ such
220 that $|\hat{\mu}_1 - \mu| \leq 2\sigma$. The analyst now uses these to compute interval $I = [\hat{\mu}_1 - \hat{\sigma}(2 + \sqrt{\ln(4n)}), \hat{\mu}_1 +$
221 $\hat{\sigma}(2 + \sqrt{\ln(4n)})]$, where $I$ is intentionally constructed to (with high probability) contain the points
222 of $\Omega(n)$ users. The analyst then passes $I$ to users in $U_2$. Users in $U_2$ respond with noisy responses
223 via independent calls to UVRR2. In UVRR2, each user clips their sample $x_i$ to the interval $I$ and
224 reports a private version $\tilde{y}_i$ using Laplace noise scaled to $|I|$.

---

**Algorithm 6** UVRR2

**Input:** $\varepsilon, i, I$
  1: User $i$ computes $x_i' \leftarrow \arg\min_{x \in I} |x - x_i|$
  2: User $i$ outputs $\tilde{y}_i \leftarrow x_i' + \mathsf{Lap}(|I|/\varepsilon)$
**Output:** Private version of user's point clipped to $I$

---

225 The average of these $\tilde{y}_i$ then approximates $\mu$. We formalize this in the following lemma, which proves
226 our main result.

**Lemma 3.5.** *Conditioned on the success of the previous lemmas, with probability at least $1 - \beta$,*
$|\hat{\mu}_2 - \mu| = O\left(\frac{\sigma}{\varepsilon}\sqrt{\frac{\log(1/\beta)\log(n)}{n}}\right).$

*Proof.* There are two sources of error in the analyst's estimate $\hat{\mu}_2 = \frac{2}{n}\sum_i \tilde{y}_i$: error from the unnoised $x_i'$s and error from noise in $\tilde{y}_i$s. Specifically, recalling that $|U_2| = n/2$, we can decompose $\hat{\mu}_2$ as

$$\hat{\mu}_2 = \frac{2}{n}\sum_i \tilde{y}_i = \frac{2}{n}\sum_i (x_i' + \eta_i)$$

where each $\eta_i \sim_{i.i.d.} \mathsf{Lap}(|I|/\varepsilon)$ and $|I| = 2\hat{\sigma}(2 + \sqrt{\ln(4n)})$.

First, using $n > 4\ln(3/\beta)$ by concentration of independent Laplace random variables (see e.g. Lemma 2.8 in Chan et al. [3]) with probability at least $1 - \beta/3$,

$$\left|\frac{2}{n}\sum_i \eta_i\right| \le \frac{4|I|}{\varepsilon}\sqrt{\frac{2\ln(3/\beta)}{n}} \le \frac{8\hat{\sigma}(2 + \sqrt{\ln(4n)})}{\varepsilon}\sqrt{\frac{2\ln(3/\beta)}{n}} = O\left(\frac{\hat{\sigma}}{\varepsilon}\sqrt{\frac{\log(1/\beta)\log(n)}{n}}\right).$$

This bounds the contribution of Laplace noise to overall error.

It remains to bound $|\frac{2}{n}\sum_i x_i' - \mu|$. Let $V$ denote the set of users with $x_i \in I$ and $V^c$ denote the set of users with $x_i \notin I$. First, by a Gaussian tail bound, for each user $i$, $\mathbb{P}\left[|x_i - \mu| \ge \sigma\sqrt{\ln(4n)}\right] \le 1/\sqrt{n}$. Then by a Chernoff bound

$$\mathbb{P}\left[|V^c| > \left(1 + \sqrt{\frac{6\ln(3/\beta)}{n^{3/2}}}\right)\sqrt{n}\right] \le \beta/3$$

and using $n \ge (6\ln(2/\beta))^{2/3}$ we get $\sqrt{\frac{6\ln(3/\beta)}{n^{3/2}}} \le 1$, so with probability at least $1 - \beta/3$, $|V^c| \le 2\sqrt{n}$. Thus

$$\frac{2}{n}\sum_{i \in V^c} |x_i' - \mu| \le \frac{2}{n}(|V^c| \cdot |I|) \le \frac{6\hat{\sigma}(2 + \sqrt{\ln(4n)})}{\sqrt{n}} = O\left(\frac{\hat{\sigma}\sqrt{\log(n)}}{\sqrt{n}}\right).$$

This bounds the contribution of error from the (unprivatized) data of users in $V^c$. Let $V$ denote the set of users in $U_2$ with points in $I$. We bound the error contributed by users in $V$ in a similar way. Users in $V$ have $x_i' = x_i$, so by a Chernoff bound on (shifted) $[0, |I|]$-bounded random variables, with probability at least $1 - \beta/3$

$$\frac{2}{n}\sum_{i \in V^c} |x_i' - \mu| = \frac{2}{n}\sum_{i \in V^c} |x_i - \mu| \le |I|\sqrt{\frac{2\ln(6/\beta)}{n}} \le \hat{\sigma}(2 + \sqrt{\ln(4n)})\sqrt{\frac{2\ln(6/\beta)}{n}} = O\left(\frac{\hat{\sigma}\sqrt{\log(1/\beta)\log(n)}}{\sqrt{n}}\right).$$

Putting these three bounds together, we get

$$\left|\frac{2}{n}\sum_i \tilde{y}_i - \mu\right| \le \frac{2}{n}\sum_i |x_i' + \eta_i - \mu|$$

$$\le \frac{2}{n}\sum_i |x_i' - \mu| + \frac{2}{n}\sum_i |\eta_i|$$

$$= \frac{2}{n}\sum_{i \in V} |x_i' - \mu| + \frac{2}{n}\sum_{i \in V^c} |x_i' - \mu| + \frac{2}{n}\sum_i |\eta_i|$$

$$= O\left(\frac{\sigma}{\varepsilon}\sqrt{\frac{\log(n)\log(1/\beta)}{n}}\right)$$

where the last step uses $\hat{\sigma} \in [\sigma, 8\sigma]$ from Lemma 3.3. $\qquad\square$

# 4   Proofs from Section 4.2

We start with pseudocode for 1ROUNDUVGAUSSTIMATE.

**Algorithm 7** 1ROUNDUVGAUSSTIMATE

**Input:** $\varepsilon, k_1, k_2, \mathcal{L}_1, n, R, \sigma, U_1, U_2,$
 1: Analyst computes $\rho \leftarrow \lceil \sqrt{2\ln(2\sqrt{n})} + 6 \rceil$
 2: **for** $j \in \mathcal{L}_1$ **do**
 3:     **for** user $i \in U_1^j$ **do**
 4:         User $i$ outputs $\tilde{y}_i \leftarrow \text{RR1}(\varepsilon, i, j)$
 5:     **end for**
 6: **end for**
 7: **for** $j_1 \in \mathcal{L}_1$ **do**
 8:     **for** $j_2 \in R_{j_1}$ **do**
 9:         **for** user $i \in U_2^{j_1, j_2}$ **do**
10:             User $i$ outputs $\tilde{y}_i \leftarrow \text{1ROUNDUVRR2}(\varepsilon, i, j_1, j_2, \rho, S)$
11:         **end for**
12:     **end for**
13: **end for**                                                ▷ End of round 1
14: Analyst computes $\hat{H}_1 \leftarrow \text{AGG1}(\varepsilon, k_1, \mathcal{L}_1, U_1)$
15: Analyst computes $\hat{\sigma} \leftarrow \text{ESTVAR}(\beta, \varepsilon, \hat{H}_1, k_1, \mathcal{L})$
16: Analyst computes $j_1 \leftarrow \log(\hat{\sigma})$
17: Analyst computes $\hat{H}_2 \leftarrow \text{KVAGG1}(\varepsilon, k_1, \mathcal{L}_1, U_1)$
18: Analyst computes $\hat{\mu}_1 \leftarrow \text{ESTMEAN}(\beta, \varepsilon, \hat{H}_2, k_1, \mathcal{L}_1)$
19: Analyst computes $j_2 \leftarrow \arg\min_{j \in R_{j_1}} \left( \min_{s \in S(j_1, j)} |s - \hat{\mu}_1| \right)$
20: Analyst computes $s^* \leftarrow \min_{s \in S(j_1, j_2)} |s - \hat{\mu}_1|$
21: Analyst outputs $\hat{\mu}_2 \leftarrow s^* + \frac{1}{k_2} \sum_{i \in U_2^{j_1, j_2}} \tilde{y}_i$
**Output:** Analyst estimate $\hat{\mu}_2$ of $\mu$

248 1ROUNDUVGAUSSTIMATE's privacy guarantee follows from the same analysis of randomized
249 response and Laplace noise as for UVGAUSSTIMATE, so we omit its proof.

250 **Theorem 4.1.** 1ROUNDUVGAUSSTIMATE *satisfies* $(\varepsilon, 0)$-*local differentially privacy for* $x_1, \ldots, x_n$.

251 We define $k_1, \mathcal{L}_1,$ and $U_1$, as in UVGAUSSTIMATE and skip the analysis of 1ROUNDUVGAUSSTI-
252 MATE's treatment of users in $U_1$ as it is identical to that of UVGAUSSTIMATE. We recall its collected
253 guarantee:

254 **Lemma 4.2.** *With probability at least* $1 - \beta$, $\hat{\sigma} \in [\sigma, 8\sigma]$ *and* $|\hat{\mu}_1 - \mu| \le 2\sigma$.

255 We again define $R$ and $S$ for $U_2$, albeit with a few modifications. First, we let $\rho = \lceil \sqrt{\ln(4n)} + 6 \rceil$
256 for neatness. Then, recalling from Section 3 that $\mathcal{L}_1$ ranges over possible values of $\log(\sigma)$, for
257 each $j_a \in \mathcal{L}_1$ we define $R_{j_a} = \{2^{j_a}, 2 \cdot 2^{j_a}, \ldots, \rho \cdot 2^{j_a}\}$. Next, for each $j_a \in \mathcal{L}_1$ and $j_b \in R_{j_a}$, we
258 define $S(j_a, j_b) = \{j_b + b\rho 2^{j_a} \mid b \in \mathbb{Z}\}$. Finally, we split $U_2$ into $L_1 \cdot \rho$ subgroups $U_2^{j_a, j_b}$ of size
259 $k_2 = \Omega\left( \frac{n}{\log\left(\frac{\sigma_{\max}}{\sigma_{\min}} + 1\right)\sqrt{\log(n)}} \right)$ for each $j_a \in \mathcal{L}_1$ and $j_b \in R_{j_a}$. As in 1ROUNDKVGAUSSTIMATE, we
260 parallelize over these subgroups to simulate the second round of UVGAUSSTIMATE for different
261 values of $(j_a, j_b)$.

262 In each subgroup $U_2^{j_a, j_b}$, each user $i$ computes the nearest element $s_i \in S(j_a, j_b)$ to $x_i$, $s_i =$
263 $\arg\min_{s \in S(j_a, j_b)} |x_i - s|$ and outputs $x_i - s_i$ plus Laplace noise in 1ROUNDUVRR2. The analyst then
264 uses estimates $j_1 = \lceil \log(\hat{\sigma}) \rceil$ and $\hat{\mu}_1$ from $U_1$ to compute $j_2 = \arg\min_{j \in R_{j_1}} \left( \min_{z \in S(j_1, j)} |z - \hat{\mu}_1| \right)$.
265 Finally, the analyst aggregates randomized responses from group $U_2^{j_1, \hat{\mu}_2}$ into an estimate $\hat{\mu}_2$.

**Algorithm 8** 1ROUNDUVRR2

**Input:** $\varepsilon, i, j_1, j_2, \rho, S$
 1: User $i$ computes $s_i \leftarrow \min_{s \in S(j_1, j_2)} |s - x_i|$
 2: User $i$ computes $y_i \leftarrow x_i - s_i$
 3: User $i$ outputs $\tilde{y}_i \leftarrow y_i + \text{Lap}\left(2\rho 2^{j_1}/\varepsilon\right)$
**Output:** Private version of user's point $x_i$

As in 1ROUNDKVGAUSSTIMATE, we start with a concentration result for each $U_2^{j_1,j_2}$. Since its proof is similar to that of Lemma 2.3, we omit it.

**Lemma 4.3.** *With probability at least* $1 - \beta$, *for all* $j_1 \in \mathcal{L}_1$ *and* $j_2 \in R_{j_1}$, *group* $U_2^{j_1,j_2}$ *contains* $\leq 2\sqrt{k_2}$ *users* $i$ *such that* $|x_i - \mu| > \sigma\sqrt{\ln(4n)}$.

In combination with the previous lemmas, this enables us to prove our final accuracy result.

**Lemma 4.4.** *Conditioned on the success of the previous lemmas, with probability at least* $1 - \beta$, 1ROUNDUVGAUSSTIMATE *outputs* $\hat{\mu}_2$ *such that*

$$|\hat{\mu}_2 - \mu| = O\left(\frac{\sigma}{\varepsilon}\sqrt{\frac{\log\left(\frac{\sigma_{\max}}{\sigma_{\min}} + 1\right)\log(1/\beta)\log^{3/2}(n)}{n}}\right).$$

*Proof.* By Lemma 4.2, $\hat{\sigma} \in [\sigma, 8\sigma]$ and $|\hat{\mu}_1 - \mu| \leq 2\sigma$. Since $j_1 = \log(\hat{\sigma}) \in \mathcal{L}_1$ and $j_2 = \arg\min_{j \in R_{j_1}} \left(\min_{s \in S(j_1,j)} |s - \hat{\mu}_1|\right)$, by the definition of $s^* \in S(j_1, j_2)$, $|s^* - \hat{\mu}_1| \leq 0.5\hat{\sigma} < 4\sigma$. Thus $|s^* - \mu| < 6\sigma$.

Consider group $U_2^{j_1,j_2}$. By Lemma 4.3 at most $2\sqrt{k_2}$ users $i \in U_2^{j_1,j_2}$ have $|x_i - \mu| > \sigma\sqrt{\ln(4n)}$. Thus by $|s^* - \mu| < 6\sigma$ and the fact that any two points in $S(j_1, j_2)$ are at least $\hat{\sigma}\rho \geq \sigma(6 + \sqrt{\ln(4n)})$ far apart, we get that at least $k_2 - 2\sqrt{k_2}$ users $i \in U_2^{j_1,j_2}$ set $s_i = s^*$ in their run of 1ROUNDUVRR2. Denote this subset of users by $V$, and denote by $V^c$ the set of users $i \in U_2^{j_1,j_2}$ such that $s_i \neq s^*$, and for each user $i \in U_2$ let $y_i = x_i - s_i$.

Let $f(x) = \frac{1}{\sigma\sqrt{2\pi}}\exp(-(x-\mu)^2/2\sigma^2)$, the density for $N(\mu, \sigma^2)$. Then

$$\int_\infty^\infty (x-\mu)f(x)dx = \int_{-\infty}^{s^*-\rho\hat{\sigma}} (x-\mu)f(x)dx + \int_{s^*-\rho\hat{\sigma}}^{s^*+\rho\hat{\sigma}} (x-\mu)f(x)dx + \int_{s^*+\rho\hat{\sigma}}^\infty (x-\mu)f(x)dx. \quad (4)$$

Let $g(x) = -\frac{\sigma}{\sqrt{2\pi}}\exp(-(x-\mu)^2/2\sigma^2)$, the antiderivative of $(x-\mu)f(x)$. Then

$$\left|\int_{-\infty}^{s^*-\rho\hat{\sigma}} (x-\mu)f(x)dx\right| = \left|g(s^* - \rho\hat{\sigma}) - \lim_{b\to-\infty} g(b)\right|$$

$$= \left|\frac{\sigma}{\sqrt{2\pi}} \cdot \exp\left(-\frac{(s^* - \rho\hat{\sigma} - \mu)^2}{2\sigma^2}\right)\right|$$

$$\leq \left|\frac{\sigma}{\sqrt{2\pi}} \cdot \exp\left(-\frac{([6-\rho]\sigma)^2}{2\sigma^2}\right)\right|$$

$$\leq \left|\frac{\sigma}{\sqrt{2\pi}} \cdot \exp\left(-\frac{[6-\rho]^2}{2}\right)\right|$$

$$< \frac{\sigma}{\sqrt{2\pi}} \cdot \exp(-\ln(2\sqrt{n}))$$

$$< \frac{\sigma}{\sqrt{n}}$$

where the first inequality uses $\hat{\sigma} \geq \sigma$ and $|s^* - \mu| < 6\sigma$. Similar logic implies $\left|\int_{s^*+\rho\hat{\sigma}}^\infty (x-\mu)f(x)dx\right| \leq \sigma/\sqrt{n}$ as well. Therefore by Equation 4 and $\int_{-\infty}^\infty (x-\mu)f(x)dx = 0$,

$$\left|\int_{s^*-\rho\hat{\sigma}}^{s^*+\rho\hat{\sigma}} (x-\mu)f(x)dx\right| \leq 2\sigma/\sqrt{n}$$

so by $\mathbb{E}\left[x_i \cdot \mathbb{1}(i \in V)\right] = \int_{s^*-\rho\hat{\sigma}}^{s^*+\rho\hat{\sigma}} xf(x)dx$, we get

$$\left|\mathbb{E}\left[x_i \cdot \mathbb{1}(i \in V)\right] - \mu\int_{s^*-\rho\hat{\sigma}}^{s^*+\rho\hat{\sigma}} f(x)dx\right| \leq 2\sigma/\sqrt{n}.$$

Since $\mathbb{E}\left[x_i \cdot \mathbb{1}(i \in V)\right]/\mathbb{P}\left[i \in V\right] = \mathbb{E}\left[x_i \mid i \in V\right]$ and $\mathbb{P}\left[i \in V\right] = \int_{s^*-\rho\hat{\sigma}}^{s^*+\rho\hat{\sigma}} f(x)dx$, this means

$$\left|\mathbb{E}\left[x_i \mid i \in V\right] - \mu\right| \leq 2\sigma/\sqrt{n}.$$

287  By $y_i = x_i - s^*$ for $i \in V$,

$$|\mathbb{E}[y_i \mid i \in V] - (\mu - s^*)| \leq 2\sigma/\sqrt{n}.$$

288  We can therefore decompose

$$\left| \frac{1}{k_2} \sum_{i \in U_2^{j_1,j_2}} y_i - (\mu - s^*) \right| \leq \left| \frac{1}{k_2} \sum_{i \in V} (y_i - (\mu - s^*)) \right| + \left| \frac{1}{k_2} \sum_{i \in V^c} (y_i - (\mu - s^*)) \right|$$

$$\leq \left[ \frac{2\sigma}{\sqrt{n}} + \rho\hat{\sigma}\sqrt{\frac{2\log(4/\beta)}{k_2}} \right] + \frac{2\rho\hat{\sigma}}{\sqrt{k_2}}$$

$$= O\left( \sigma\sqrt{\frac{\log(1/\beta)\log(n)}{k_2}} \right)$$

289  where the the first inequality uses a (with probability at least $1 - \beta/2$) Chernoff bound on $\{y_i \mid i \in V\}$
290  concentrating around $\mathbb{E}[y_i \mid i \in V]$ as well as $|V^c| \leq 2\sqrt{k_2}$, and the last step uses $\hat{\sigma} \in [\sigma, 8\sigma]$.

291  Next, since we can decompose

$$\frac{1}{k_2} \sum_{i \in U_2^{j_1,j_2}} \tilde{y}_i = \frac{1}{k_2} \sum_{i \in U_2^{j_1,j_2}} y_i + \frac{1}{k_2} \sum_{i \in U_2^{j_1,j_2}} \eta_i$$

292  where each $\eta_i \sim \mathsf{Lap}(\rho\hat{\sigma}/\varepsilon)$, the same concentration of Laplace noise from Lemma 3.5 says that with
293  probability $1 - \beta/2$,

$$\left| \frac{1}{k_2} \sum_{i=1}^{k_2} \eta_i \right| = O\left( \frac{\rho\hat{\sigma}}{\varepsilon}\sqrt{\frac{\log(1/\beta)}{k_2}} \right) = O\left( \frac{\sigma}{\varepsilon}\sqrt{\frac{\log(1/\beta)\log(n)}{k_2}} \right).$$

294  Combining with the bound above and substituting in $k_2 = \Omega\left( \frac{n}{\log\left( \frac{\sigma_{\max}}{\sigma_{\min}}+1 \right)\sqrt{\log(n)}} \right)$,

$$\left| \frac{1}{k_2} \sum_{i \in U_2^{j_1,j_2}} \tilde{y}_i - (\mu - s^*) \right| = O\left( \frac{\sigma}{\varepsilon}\sqrt{\frac{\log\left( \frac{\sigma_{\max}}{\sigma_{\min}} + 1 \right)\log(1/\beta)\log^{3/2}(n)}{n}} \right).$$

295  The claim then follows from $\hat{\mu}_2 = s^* + \frac{1}{k_2}\sum_{i \in U_2^{j_1,j_2}} \tilde{y}_i$.  $\square$

# 5   Proofs from Section 5

297  For completeness, we start with the formal notion of sequential interactivity used by Duchi et al. [5],
298  which requires that the set of messages $\{Y_i\}$ sent by the users satisfies the following conditional
299  independence structure: $\{X_i, Y_1, \ldots, Y_{i-1}\} \rightarrow Y_i$ and $Y_i \perp X_j \mid \{X_i, Y_1, \ldots, Y_{i-1}\}$ for $j \neq i$. Our
300  notion of sequential interactivity — where each user only sends one message — is a specific case of
301  this general definition. Our upper bounds all meet this specific requirement, while our lower bound
302  meets the general one.

303  We start by defining an instance $\mathsf{Estimate}(n, M, \sigma)$. Here, a protocol receives $n$ samples from a
304  $N(\mu, \sigma^2)$ distribution where $\sigma$ is known, $\mu \in [0, M]$, and the goal is to estimate $\mu$. Next, define
305  uniform random variable $V \sim_U \{0, 1\}$. Consider the following testing problem: for $V = v$, if $v = 0$,
306  then each user $i$ draws a sample $x_i \sim_{iid} N(0, \sigma^2)$, while if $v = 1$ then each user $i$ draws a sample
307  $x_i \sim_{iid} N(M, \sigma^2)$. The problem $\mathsf{Test}(n, M, \sigma)$ is to recover $v$ from $x_1, \ldots, x_n$. We say protocol
308  $\mathcal{A}$ $(\alpha, \beta)$-solves $\mathsf{Estimate}(n, M, \sigma)$ if, with probability at least $1 - \beta$, $\mathcal{A}(\mathsf{Estimate}(n, M, \sigma)) = \hat{\mu}$
309  such that $|\hat{\mu} - \mu| < \alpha$. We will say that an algorithm $\mathcal{A}$ $\beta$-solves $\mathsf{Test}(n, M, \sigma)$ if, with probability at
310  least $1 - \beta$, $\mathcal{A}(\mathsf{Test}(n, M, \sigma)) = v$. Formally, $\mathsf{Test}(n, M, \sigma)$ is no harder than $\mathsf{Estimate}(n, M, \sigma)$.

311  **Lemma 5.1.** *If there exists a sequentially interactive and $(\varepsilon, \delta)$-locally private protocol $\mathcal{A}$ that*
312  *$(M/2, \beta)$-solves $\mathsf{Estimate}(n, M, \sigma)$, then there exists a sequentially interactive and $(\varepsilon, \delta)$-locally*
313  *private protocol $\mathcal{A}'$ that $\beta$-solves $\mathsf{Test}(n, M, \sigma)$.*

*Proof.* Let $x_1, \ldots, x_n$ be the samples from an instance of $\mathsf{Test}\,(n, M, \sigma)$. We define $\mathcal{A}'$ to run $\mathcal{A}(x_1, \ldots, x_n)$ and then output $\arg\min_{\hat{\mu} \in \{0, M\}} |\mathcal{A}(x_1, \ldots, x_n) - \hat{\mu}|$. Since $\mathcal{A}$ $(M/2, \beta)$-solves $\mathsf{Estimate}\,(n, M, \sigma)$, with probability at least $1 - \beta$, $|\mathcal{A}(x_1, \ldots, x_n) - \mu| < M/2$. Thus with probability at least $1 - \beta$, $\mathcal{A}'(x_1, \ldots, x_n) = v$. Thus $\mathcal{A}'$ $\beta$-solves $\mathsf{Test}\,(n, M, \sigma)$. As $\mathcal{A}'$ interacted with $x_1, \ldots, x_n$ only through $(\varepsilon, \delta)$-locally private $\mathcal{A}$, by preservation of differential privacy under postprocessing, $\mathcal{A}'$ is $(\varepsilon, \delta)$-locally private as well. Similar logic implies that $\mathcal{A}'$ is also sequentially interactive. $\qquad\square$

We now extend this result to $(\varepsilon, \delta)$-locally private protocols using results from both Bun et al. [2] and Cheu et al. [4][1].

**Lemma 5.2.** *Let* $\delta < \min\left(\frac{\epsilon\beta}{48 n \ln(2n/\beta)}, \frac{\beta}{16 n \ln(n/\beta) e^{7\varepsilon}}\right)$, $\varepsilon > 0$, *and suppose that* $\mathcal{A}$ *is a sequentially interactive and* $(\varepsilon, \delta)$-*locally private protocol. If* $\mathcal{A}$ $\beta$-*solves* $\mathsf{Test}\,(n, M, \sigma)$, *then there exists a sequentially interactive* $(10\varepsilon, 0)$-*locally private* $\mathcal{A}'$ *that* $4\beta$-*solves* $\mathsf{Test}\,(n, M, \sigma)$.

*Proof.* Our analysis splits into two cases depending on $\epsilon$.

Case 1: $\varepsilon \le 1/4$. In this case, we use a result from Bun et al. [2], included here for completeness.

**Fact 5.3** (Theorem 6.1 in Bun et al. [2] (restated))**.** *Given* $\varepsilon \le 1/4$ *and* $\delta < \epsilon\beta/48 n \ln(2n/\beta)$, *there exists a* $(10\varepsilon, 0)$-*locally private algorithm* $\mathcal{A}'$ *such that for every database* $U = \{x_1, \ldots, x_n\}$, $d_{TV}(\mathcal{A}(U), \mathcal{A}'(U)) \le \beta$, *where* $d_{TV}$ *denotes total variation distance.*

Thus, denoting by $E_{\mathcal{A}}$ the event where $\mathcal{A}$ recovers the correct $v$ on $\mathsf{Test}\,(n, M, \sigma)$ and $E_{\mathcal{A}'}$ the event where $\mathcal{A}'$ recovers the correct $v$ on $\mathsf{Test}\,(n, M, \sigma)$, $|\mathbb{P}\left[E_{\mathcal{A}}\right] - \mathbb{P}\left[E_{\mathcal{A}'}\right]| \le \beta$, where the probabilities are respectively over $\mathcal{A}$ and $\mathcal{A}'$. Thus since $\mathcal{A}$ $\beta$-solves $\mathsf{Test}\,(n, M, \sigma)$, it follows that $\mathcal{A}'$ $2\beta$-solves (and thus also $4\beta$-solves) $\mathsf{Test}\,(n, M, \sigma)$.

Case 2: $\varepsilon > 1/4$. In this case we use a result from Cheu et al. [4][2]

**Fact 5.4** (Theorem A.1 in Cheu et al. [4] (restated))**.** *Given* $\varepsilon > 1/4$ *and* $\delta < \frac{\beta}{16 n \ln(n/\beta) e^{7\varepsilon}}$, *there exists an* $(8\varepsilon, 0)$-*locally private protocol* $\mathcal{A}'$ *such that* $\mathcal{A}'$ $4\beta$-*solves* $\mathsf{Test}\,(n, M, \sigma)$.

$\qquad\square$

Finally, we prove that $\mathsf{Test}$ is hard for $(\varepsilon, 0)$-locally private protocols. At a high level, we prove this result by viewing $\mathsf{Test}$ as a Markov chain $V \to$ data $X \to$ outputs $Y \to$ answer $Z$. We bound the mutual information $I(V; Z)$ by a function of $M, \sigma$, and $I(X; Y)$ using a strong data processing inequality for Gaussian distributions (see Section 4.1 in Braverman et al. [1] or Raginsky [8] for details; a primer on information theory appears in the last section). We further bound $I(X; Y)$ using existing tools from the privacy literature [5]. The resulting upper bound on $I(V; Z)$ enables us to lower bound the probability of an incorrect answer $Z$.

**Lemma 5.5.** *Suppose* $M \le \sigma / [4(e^\varepsilon - 1)\sqrt{2nc}]$, *where* $c$ *is an absolute constant. For any sequentially interactive and* $(\varepsilon, 0)$-*locally private protocol* $\mathcal{A}$ *that* $\beta$-*solves* $\mathsf{Test}\,(n, M, \sigma)$, $\beta \ge 1/4$.

*Proof.* We may express any sequentially interactive $(\varepsilon, 0)$-locally private protocol $\mathcal{A}$ that $\beta$-solves $\mathsf{Test}\,(n, M, \sigma)$ as a Markov chain $V \to X \to Y \to Z$, where $V$ is the random variable selecting $v$, $X = (x_1, \ldots, x_n)$ is the random variable for users' i.i.d. samples, $Y = (y_1, \ldots, y_n)$ is the random variable for users' $(\varepsilon, 0)$-privatized responses, and $Z = \mathcal{A}(\mathsf{Test}\,(n, M, \sigma))$. As $V \to X \to Y \to Z$ is a Markov chain (i.e., any two random variables in the chain are conditionally independent given a random variable between them). Thus by a strong data processing inequality for two Gaussians (see e.g. Section 4.1 in Braverman et al. [1] or, for a broader treatment of strong data processing inequalities, Raginsky [8]), there exists absolute constant $c$ such that for each user $i$, $I(V; Y_i) \le$

355   $\frac{cM^2}{\sigma^2} I(X_i; Y_i)$, where $I(A;B)$ denotes the mutual information between random variables $A$ and $B$.
356   Next, since our protocol is $(\varepsilon, 0)$-locally private, by Corollary 1 from Duchi et al. [5], for each user $i$,
357   $I(X_i; Y_i) \le 4(e^\varepsilon - 1)^2$. With the equation above, we get

$$I(V; Y_i) \le \frac{4cM^2(e^\varepsilon - 1)^2}{\sigma^2}. \tag{5}$$

358   Without loss of generality, suppose $Z$ is a deterministic function of $Y$ (if $Z$ is a random function of $Y$
359   then it decomposes into a convex combination of deterministic functions of $Y$). From Markov chain
360   $V \to X \to Y \to Z$ and the (generic) data processing inequality we get

$$
\begin{aligned}
I(V; Z) \le I(V; Y_1, \ldots, Y_n) \\
= \sum_{i=1}^n I(V; Y_i \mid Y_{i-1}, \ldots Y_1) \\
\le \sum_{i=1}^n I(V, Y_{i-1}, \ldots, Y_1; Y_i) \\
= \sum_{i=1}^n \left[ I(V; Y_i) + I(Y_{i-1}, \ldots, Y_1; Y_i | V) \right] \\
= \sum_{i=1}^n I(V; Y_i)
\end{aligned}
$$

361   where the last step follows from the independence of $Y_i$ and $Y_1, \ldots, Y_{i-1}$ given $V$. Substituting in
362   Equation 5, $I(V;Z) \le \frac{4ncM^2(e^\varepsilon - 1)^2}{\sigma^2}$. Therefore by $M \le \sigma/4(e^\varepsilon - 1)\sqrt{2nc}$ we get $I(V;Z) \le 1/8$.
363   Define $P$ to be the distribution of $Z$ (over the randomness of $V$, $X$, and $Y$), and let $P_0$ and $P_1$ be the
364   distributions for $Z|V = 0$ and $Z|V = 1$ respectively. Then as $V$ is uniform, $P = (P_0 + P_1)/2$, so

$$\|P - P_0\|_1 = \|P - P_1\|_1 = \tfrac{1}{2}\|P_0 - P_1\|_1.$$

365   Moreover, by

$$
\begin{aligned}
\mathbb{P}\left[Z = V\right] &= \mathbb{P}\left[Z = 0, V = 0\right] + \mathbb{P}\left[Z = 1, V = 1\right] \\
&= \frac{1}{2}\left(P_0(0) + [1 - P_1(0)]\right) \\
&\le \frac{1}{2}\left(1 + |P_0(0) - P_1(0)|\right) \\
&= \frac{1}{2} + \frac{1}{4}\|P_0 - P_1\|_1
\end{aligned}
$$

366   we get $\mathbb{P}\left[Z = V\right] \le \frac{1}{2} + \frac{1}{4}\|P_0 - P_1\|_1$. Thus

$$
\begin{aligned}
\frac{\|P_0 - P_1\|_1^2}{8} &= \frac{1}{4}\left(\|P_0 - P\|_1^2 + \|P_1 - P\|_1^2\right) \\
&\le \frac{1}{2}\left(D_{KL}(P_0\|P) + D_{KL}(P_1\|P)\right) \\
&= I(Z; V) \le 1/8
\end{aligned}
$$

367   where the second-to-last inequality uses Pinsker's inequality. It follows that $\|P_0 - P_1\|_1 \le 1$. Substi-
368   tuting this into $\mathbb{P}\left[Z = V\right] \le \frac{1}{2} + \frac{1}{4}\|P_0 - P_1\|_1$, we get $\mathbb{P}\left[Z = V\right] \le \frac{3}{4}$. $\qquad\square$

369   We combine the preceding results to prove a general lower bound for Estimate as follows: for
370   appropriate $\varepsilon$ and $\delta$, by Lemma 5.1 any sequentially interactive and $(\frac{\varepsilon}{10}, \delta)$-locally private protocol
371   $\mathcal{A}$ that $(M/2, \frac{\beta}{4})$-solves Estimate $(n, M, \sigma)$ implies the existence of a sequentially interactive and
372   $(\frac{\varepsilon}{10}, \delta)$-locally private protocol $\mathcal{A}'$ that $\frac{\beta}{4}$-solves Test $(n, M, \sigma)$. Then, Lemma 5.2 implies the exis-
373   tence of a sequentially interactive and $(\varepsilon, 0)$-locally private protocol $\mathcal{A}''$ that $\beta$-solves Test $(n, M, \sigma)$.
374   By Lemma 5.5 any such $\mathcal{A}'$ that $\beta$-solves Test $(n, M, \sigma)$ has $\beta \ge 1/4$. Hardness for Test therefore
375   implies hardness for Estimate. We condense this reasoning into the following theorem.

**Theorem 5.6.** *Let* $\delta < \min\left(\frac{\epsilon\beta}{60n\ln(5n/2\beta)}, \frac{\beta}{16n\ln(n/\beta)e^{7\varepsilon}}\right)$, $\varepsilon > 0$, *and let* $\mathcal{A}$ *be a sequentially interactive* $(\varepsilon, \delta)$-*locally private* $(\alpha, \beta)$-*estimator for* $\mathsf{Estimate}\,(n, M, \sigma)$ *where* $M = \sigma/[4(e^\varepsilon - 1)\sqrt{2nc}]$, $c$ *is as in Lemma 5.5, and* $\beta < 1/16$. *Then* $\alpha \geq M/2 = \Omega\left(\frac{\sigma}{\varepsilon}\sqrt{\frac{1}{n}}\right)$.

In particular, Theorem 5.6 implies that our upper bounds are tight up to logarithmic factors for *any* sequentially interactive and $(\varepsilon, \delta)$-locally private protocol with sufficiently small $\delta$. Using recent subsequent work [7], we can also extend this result to the fully interactive setting, as shown in the next section.

## 5.1 Extension to Fully Interactive Lower Bound

The following result, proven in subsequent work by Joseph et al. [7] also relying on the work of Braverman et al. [1], gives a general lower bound for locally private simple hypothesis testing problems like Test.

**Lemma 5.7** (Theorem 5.3 in Joseph et al. [7]). *For* $\varepsilon > 0$ *and* $\delta < \min\left(\frac{\varepsilon^3\alpha^2}{48n\ln(2n/\beta)}, \frac{\varepsilon^2\alpha^2}{64n\ln(n/\beta)e^{7\varepsilon}}\right)$, *any* $(\varepsilon, \delta)$-*locally private simple hypothesis testing protocol distinguishing between distributions* $P_0$ *and* $P_1$ *with probability at least* $2/3$ *requires* $n = \Omega\left(\frac{1}{\varepsilon^2\|P_0 - P_1\|_{TV}^2}\right)$ *samples.*

Since in general $D_{KL}(N(\mu_1, \sigma^2)\|N(\mu_2, \sigma^2)) \leq \left[\frac{\mu_1 - \mu_2}{\sigma}\right]^2$, in the setting of $\mathsf{Test}\,(n, M, \sigma)$ we are distinguishing between $P_0 = N(0, \sigma^2)$ and $P_1 = N(M, \sigma^2)$ and get $D_{KL}(P_0\|P_1) = O\left(\frac{M^2}{\sigma^2}\right)$. Pinsker's inequality then implies $\|P_0 - P_1\|_{TV}^2 = O\left(\frac{M^2}{\sigma^2}\right)$. Substituting this into Lemma 5.7, we get that distinguishing $P_0$ and $P_1$ with constant probability and $n$ samples requires $M = \Omega\left(\frac{\sigma}{\varepsilon\sqrt{n}}\right)$. Thus, for appropriately small $\delta$, any $(\varepsilon, \delta)$-locally private protocol that $(\alpha, \beta)$-solves $\mathsf{Estimate}\,(n, M, \sigma)$ has $\alpha = \Omega(M) = \Omega\left(\frac{\sigma}{\varepsilon\sqrt{n}}\right)$.

# 6 Information Theory Overview

We briefly review some standard facts and definitions from information theory, starting with entropy.

**Definition 6.1.** *The* entropy $H(X)$ *of a random variable* $X$ *is*

$$H(X) = \sum_x \mathbb{P}\left[X = x\right]\ln\left(\frac{1}{\mathbb{P}[X=x]}\right),$$

*and the* conditional entropy $H(X|Y)$ *of random variable* $X$ *conditioned on random variable* $Y$ *is*

$$H(X|Y) = \mathbb{E}_y[H(X|Y = y)].$$

Next, we can use entropy to define the mutual information between two random variables. Mutual information between random variables $X$ and $Y$ is roughly the amount by which conditioning on $Y$ reduces the entropy of $X$ (and vice-versa).

**Definition 6.2.** *The* mutual information $I(X; Y)$ *between two random variables* $X$ *and* $Y$ *is*

$$I(X; Y) = H(X) - H(X|Y) = H(Y) - H(Y|X),$$

*and the* conditional mutual information $I(X; Y|Z)$ *between* $X$ *and* $Y$ *given* $Z$ *is*

$$I(X; Y|Z) = H(X|Z) - H(X|Y, Z) = H(Y|Z) - H(Y|X, Z).$$

We also define the related notion of KL-divergence.

**Definition 6.3.** *The* Kullback-Leibler divergence $D_{KL}(X\|Y)$ *between two random variables* $X$ *and* $Y$ *is*

$$D_{KL}(X\|Y) = \sum_x \mathbb{P}\left[X = x\right]\ln\left(\frac{\mathbb{P}\left[X = x\right]}{\mathbb{P}\left[Y = x\right]}\right),$$

*where we often abuse notation and let* $X$ *and* $Y$ *denote the distributions associated with* $X$ *and* $Y$.

KL divergence connects to mutual information as follows.

**Fact 6.4.** *For random variables $X$, $Y$, and $Z$,*

$$I(X;Y|Z) = \mathbb{E}_{x,z}\left[D_{KL}\left((Y|X=x, Z=z)\|(Y|Z=z)\right)\right].$$

Finally, we will also use the following connection between KL divergence and $\|\cdot\|_1$ distance.

**Lemma 6.5** (Pinsker's inequality). *For random variables $X$ and $Y$,*

$$\|X-Y\|_1 \le \sqrt{2D_{KL}(X\|Y)}.$$

## Footnotes

[1]Both of these results are stated for noninteractive protocols, it is straightforward to see that their techniques carry over to sequentially interactive protocols. This is because both results rely on transforming a single user call to an $(\varepsilon, \delta)$-local randomizer into calls to an $(O(\varepsilon), 0)$-local randomizer. Since users in sequentially interactive protocols still only make a single call to a local randomizer, we can apply the same transformations to each single user call and obtain an $(O(\varepsilon), 0)$-locally private sequentially interactive protocol.

[2] Cheu et al. [4] originally state their result for $\varepsilon > 2/3$, but mildly strengthening their assumed upper bound on $\delta$ from $\delta < \frac{\beta}{8 n \ln(n/\beta) e^{6\varepsilon}}$ to $\delta < \frac{\beta}{16 n \ln(n/\beta) e^{7\varepsilon}}$ yields the result here.