[Reviews · NeurIPS 2019]

Reviewer 1



This paper considers the problem of estimation the mean mu of a univariate Gaussian under local privacy (LDP) constraints. This work improves on previously known bounds in both the minimax rate (by shaving some logarithmic factors and achieving near-optimal rate), the number of rounds of adaptivity (how many times must the center interact with any individual user), and the guarantee itself (pure locally privacy instead of the weaker approximate local privacy). The authors provide results in two different settings: - known variance (non-adaptive and one-round-of-adaptivity protocols) - unknown variance (non-adaptive and one-round-of-adaptivity protocols) The techniques are interesting, especially the idea of first estimating the mean "bit by bit" (via their "mod 4") trick). The introduction is very clearly written, and provides a good overview (with one big caveat -- see below, about a paper of Duchi and Rogers). I am, however, less convinced by the rest of the paper: while the high-level discussion of the proofs are roughly OK, there is a constant referring to subroutines not introduced or barely discussed (KVRR1,KVRR2,KVAGG1.,KVAGG2...). I would strongly recommend an overhaul of the writeup, to change this (details below). 1. In the preliminaries, remove Definition 2.1. It is irrelevant and confusing (and takes space), as you never use global DP. 2. you give the pseudocode of Algorithms 1 and 2. This is good, but useless, as you don't give any pseudocode or formal description of *all* the subroutines it invokes. What is the point then, just showing there is pseudocode, but not allowing anyone to actually implement it? I'd recommend including these subroutines in the paper, or remove the pseudocode. 3. similarly in the proofs: you refer to those subroutines a *lot*, but never give enough details for someone not familiar with the literature to begin with. 4. remove the lower bound section entirely: indeed, all the lower bounds (unless I am mistaken) are entirely subsumed by Corollary 5 (arXiv version: https://arxiv.org/abs/1902.00582; for d=1) of Duchi-Rogers'19 (COLT'19). Since the techniques are different, it may be good to keep a paragraph pointing to the supplementary material for a different proof using the SDPI as you do, but it should not take one full page of the submission (esp. since Lemma 5.1 is trivial, and Lemma 5.2 is folklore) Beyond these "big" comments, a few more: - Discuss the Duchi-Rogers'19 (COLT'19) in the introduction. - In all your statements, correct the (IMO, bad) way you define the sufficient condition on n: "n= Omega(f(n))" (i.e., n appears in both the LHS and RHS) is not satisfying. (Theorems 3.1, 3.2, 4.1...) - In Theorem 1.1 and 1.2, drop the "alpha" introduced for "neatness": (i) it doesn't help, as it hides the important dependencies in the parameters, and (ii) it conflicts with your use of alpha in Table 1. - Theorem 1.3: I assume you meant little-Oh (o(alpha))? Otherwise, this seems to contradict the 2-round protocol of Theorem 1.1. - Section 2: " a single i.i.d. draw" doesn't really make sense. What you mean is clear, but you should rephrase it. - Section 3.1.1: Discuss/give intuition as to why you take it mod 4 and not, say, mod 2 (or 3) - Section 3.2 (ll. 163 and 167): give some intuition as to why this is sqrt(log n). Is it because of a tradeoff you are trying to optimize? If so, which one? Question: your algorithms only appear to use Gaussian concentration. Would they work for any subgaussian distribution? (other question, related: how robust are you algorithms to model misspecification, say if the distribution is only approximately Gaussian?) UPDATE: I have read the authors' response. Please, make sure to also take into account all the other comments (e.g., intuition about the sqrt(log n), about why the modulo 4, etc.) when revising the paper.

Reviewer 2



The paper studies the problem of Gaussian estimation in the local model of differential privacy. In particular, it achieves accuracy bounds that are tight under certain conditions on the mean of the underlying distribution. They also reduce the number of interaction from linear in the number of users to n. These are nice results at first glance; however, I have some reservations against accepting the paper in its present state. I list them in more details below and hope that the authors would find my comments constructive. Since this is mainly a theoretical paper, I have tried to put it at the same footing as I would expect from a theoretical paper. Theorem 1.2 sounds fishy. There is no dependence on \sigma_{min} or \sigma_{max}. My intuition is that there is should be a \log dependence on the ratio of two. I tried looking for it in the proof, but could not find where the authors have erred. Can the author provide an intuition why this is the case? For me, if there is no dependence on accuracy on the bound on variance, it is almost the same bound as in the case of Theorem 1.1 barring a factor of \sqrt{\log n}. What would be the situation when \sigma_min is exponentially small in n and \sigma_max is a polynomial. This can surely happen. Figure 1 mentions that R is an upper bound on both mean and variance which the authors mention is exponential in n. If that is the case, then their result on unknown non-adaptive would cancel the factor of n in the \sqrt. Maybe, I am missing something, but does that not make the result trivial? Perhaps, there is an accuracy interaction trade-off. The lower bound is only true when mean is of an order of the accuracy loss. So I think there is a gap between the lower and upper bound and it is misleading to say that they have tight bounds. An argument along this line would be helpful in the presentation of the paper. Related works. I do not get the result in comparison to that in the central model. The authors claim that there is a roughly $\sqrt{n}$ loss in accuracy in moving to the local model of privacy; however, for large enough $n$, the authors mention that Karwa and Vadhan give a bound of $O(\sigma \sqrt{\frac{\log(1/\beta)}{n}})$; their result also gives the same bound. I am a little perplexed with the bound of the earlier result ā€” there is no dependence on \epsilon! I think that the authors have incorrectly cited the earlier papers. I would suggest the authors double-check the results in the earlier papers. Strong data processing inequality has been used previously in the context of the local model of privacy (DJW FOCS 2013 and DR COLT 2019 paper). In fact, the main claim in DJW13 uses strong data processing inequality to get a better bound than DRV10. The paper seems to claim that they are the first to use it. Further, the STOC 2016 paper on statistical estimation problems by Braverman et al. uses such inequalities and if I understand Duchi and Rogers correctly, they extend the techniques of Braverman et al. to the privacy setting. I wonder how much the current paper's technique is different from theirs. If not, I feel that the paper oversell their result. Now coming to the writing of the paper. I felt the proofs are written in a little convoluted manner and hard to follow. It is not clear what the authors want to say in lines 124-126. Unless I am mistaken, the maths says something else and the authors write something else: ā€œU_1 is split into L subgroups indexed by \mathcal Lā€. \mathcal is just a set of contiguous indices of size L. I believe they want to say that U_1 is split in to disjoint subgroup each of size k and indices are in set \mathcal L_1, \cdots, \mathcal L_k. That seems more reasonable. For the rest of the paper, I have assumed this. Disclaimer: Since the case for unknown variance should follow more or less by making \log \frac{\sigma_max}{\sigma_min} bins, I skipped Section 4 in the extended abstract. I am more curious as to why they do not incur a dependence on \sigma_min and \sigma_max. On the Supplementary material: Algorithm 2: Line 2 should have a \in \{ 0,1,2,3 \}. Details in the proof of Claim 1 are missing. The authors just mention that the proof follows by induction without even stating the base case and how the inductive argument follows. Since this is primarily a theoretical paper, these details should be spelled out clearly. I find it unsettling and make me very unsure about accepting the paper. Claim 2 assumes \hat H_1^j(M_1) < 0.52 k +\psi. Then how does line 42 follows? Claim 1 has a completely different assumption. Lemma 1.4 is the main part and its proof is very similar to that in STOC 2016 paper by Braverman et al., including the notations. Still, I did not understand where is the price of privacy coming in to play? I might be missing something but I think the authors should mention it clearly. Same could be said about Lemma 2.4. Line 81 and 86: Define \erf and \erf^{-1}. Line 143: Perhaps strikeout "then." Equation (3). Why is expectation over y_i? A line of argument should be there. A personal suggestion in improving the clarity of the paper (the authors need not take my advice). I would suggest the authors to just focus on known variance case in the extended abstract section, explain the intuition and why the previous result fails in more detail. The case for the unknown variance can be simply relegated to the appendices. This would make the paper more readable and would definitely invoke more follow-up work. Also, in the proofs, it would be helpful to give an intuition of the proof instead of putting all the maths. If the authors prefer to just put the math (which I personally do not mind), they should include every detail. I did not get the time to verify the rest of the proof. I would try to read them before the notification time in hope that it would help the authors make their paper more presentable. Post-rebuttal: I read the rebuttal provided by the authors. They did give some idea as to how to complete their proof, so that is good. On the other hand, they did not answer my questions regarding the novelty of techniques. Reading the comments of the other reviewers and their feedback, I do believe that the current write-up needs significant improvement. The authors did not even provide any promise on that front. I am raising my score to 6 on the promise that the authors would consider the reviewers' suggestion in the next version.

Reviewer 3



- I am not very familiar with this line of research works, hence I may not able to precisely evaluate the significance of these results. Overall, I think that the paper is well-written and presents relevant results. - The proofs in the Supplement seem to be correct. However, the proof sketches in the main paper are not referring to the lemmas in the Supplement. Now it leaves to readers to figure out when to apply each lemma, and what kind of conditions/guarantees are needed/obtained at each step. Also, some assumptions used in the proofs are not clearly stated, e.g., line 34 in the Supplement uses a lower bound on k, and line 85 uses a lower bound on n. - The paper reads well, and it is not hard to follow the high-level arguments. But as stated above, it is hard to understand the details and reconstruct the complete proofs as the lemmas are detached from those high-level sketches. It would be better if the authors can improve this aspect of writing. It is also worth providing more intuitions for the construction of these algorithms. - The algorithms seem non-trivial as each of them consists of a sequence of subroutines. The submission compares only the theoretical guarantees of the algorithms in [16] and the proposed ones. It may be better if the authors can explain the differences and connections between the proposed algorithms and existing ones. As for now, I don't see how novel these algorithms are. More concretely, I am not sure whether the authors are adding new ingredients to existing algorithms that are already nontrivial on their own, or creating completely new ones. - It is of interest to see the actual performance of these algorithms and compare them with their globally private or non-private alternatives. - Minor: line 255, "Testing" instead of "esting". ************** I went through the reviews and the author's rebuttal. I would like to keep my original overall score since I believe that the results are nontrivial. Yet as I mentioned in my original review, I am not very familiar with this line of work. As a result of this, I have chosen a confidence score of 3.

[Author Response · NeurIPS 2019]

We thank the reviewers for their attentive and thorough reviews! We first respond to common concerns before moving
to reviewer-specific ones below. As a general rule, we will fix all minor points/typos.

**Common concerns**: 1. *The lower bound takes up a lot of space for a result that is subsumed by Duchi-Rogers 19.*
As noted in the introduction, we agree that the lower bounds of Duchi-Rogers 19 generalize and subsume our lower
bound. However, we released the original preprint of our results (complete with the first adaptation of the SDPI lower
bound from Braverman et al.) a few months before the preprint of Duchi-Rogers 19, so we feel it is still an important
contribution. We therefore included the lower bound statement in the main body. However, we will add material in the
introduction clarifying that Duchi-Rogers subsumes our work, and will move the bulk of the lower bound section to the
appendix in the final version of the paper. As a side effect, this will free up space to add the changes promised below.

2. *The proof sketches in the main body are somewhat unclear and disconnected from the proofs in the Supplement.* We
will revisit the sketches and add explicit references connecting sketch claims and their precise claims in the Supplement.

**Reviewer 1**: 1. *Remove the definition of central differential privacy.* We will remove this.

2. *The inclusion/omission of different routines and subroutines is confusing.* We note that all pseudocode for subroutines
appears in the Supplement. Given space constraints, we felt it was clearest to include pseudocode for the main routines
(to be explicit about how the subroutines fit together) and descriptions for the simpler subroutines. We will revisit and
expand our descriptions of the subroutines and move pseudocode to the main body where appropriate.

3. *Do these techniques work for any subgaussian/near-gaussian distribution?* The bulk of our techniques should
generalize to sub/gaussian/near-gaussian distributions. A possible exception is the comparison to the Gaussian CDF (via
erf) used in the known-variance case, which may suffer given CDFs that are actually not close to Gaussian. However,
the slightly worse (but less delicate) Laplace noise method used in the unknown-variance case should perform better.

**Reviewer 2**: 1. *Where is $\sigma_{\max}/\sigma_{\min}$ in part one of Theorem 1.2?* In the corresponding formal statement, Theorem 4.1,
we assume $n = \Omega(\log(\sigma_{\max}/\sigma_{\min}))$. This is the "sufficiently large $n$" mentioned in the informal Theorem 1.2.

2. *If $R \geq \mu$, but $\sigma$ can be exponential in $n$, doesn't Gaboardi et al's bound become vacuous?* When $\mu$ (but not $\sigma$) is
exponential in $n$, their claim is not vacuous, but their round complexity is still $\Omega(n)$. We will clarify this.

3. *The lower bound only holds when the mean is of an order of the accuracy.* We agree that instance optimal lower
bounds are better. However, we think that the worst case optimality shown by our current lower bound is still useful.

4. *Where is the dependence on $\varepsilon$ in the centrally private result?* The centrally private upper bound is $O(\sigma\sqrt{\log(1/\beta)/n}+$
$\operatorname{poly}\log(1/\beta)[\varepsilon n])$. Ignoring $\log$ factors, the additional cost of central privacy over the non-private case is the second
term $O(1/[\varepsilon n])$ (lower-order in $n$) whereas local privacy costs $O(1/(\varepsilon\sqrt{n}))$ (not lower-order).

5. *Isn't DJW13 the first local privacy paper to use an SDPI approach?* DJW13 use the term SDPI in a different way.
Informally, their SDPI only controls the mutual information between a sample and its locally private output. In contrast,
our SDPI also takes advantage of a bound on the mutual information between the distribution-specifying parameter and
a sample (and is the first to do so). It is this latter interpretation that is typically the focus of the strong data processing
inequality literature. Our SDPI therefore provides strictly stronger mutual information bounds overall.

6. *I am confused by the induction in the proof of Claim 1. Also, how can you use Claim 1 in the proof of Claim 2, which*
*has a different assumption?* Claim 1 is a statement about interval $I_j$ assuming that a property holds for successive
supersets of $I_j$, $I_{j+1}, I_{j+2}, \ldots, I_{L_{\max}}$. Since $u \in I_{L_{\max}}$, the property holds for $j = L_{\max}$, the base case. We will
clarify this. For the second point, note that Claim 2 is about the maximum $j$ with $\hat{H}_1^j(M_1(j)) < 0.52k + \psi$. Thus for
$j' > j$, $\hat{H}_1^j(M_1(j')) \geq 0.52k + \psi$. This requirement on $j' > j$ is used in Claim 1 (the $M_1(j)$ should be $M_1(j')$ in the
second line of Claim 1; we will fix this). Thus we can apply Claim 1 to get $\mu \in I_j$, as written.

8. *Where is privacy used in proving Lemmas 1.4 and 2.4?* In the proof of Lemma 1.4, privacy enters when analyzing
$|T - \hat{T}|$ (lines 90-96). In the proof of Lemma 2.4, this happens in lines 150-157. We will better highlight this.

9. *In Equation (3) of the Supplement, why is it the expectation of $y_i$?* $E[y_i] = E[y]$.

**Reviewer 3**: 1. *More comparison to existing approaches would be helpful.* We will add more comparison. For example:
Gabordi et al. use a similar "binary search" approach, but they do not use our modular single round trick (which
dramatically reduces round complexity). They also use a Gaussian noise approach for known variance (less accurate
than our use of Braverman et al's CDF-based approach). Karwa and Vadhan employ an approach based on the histogram
that crucially relies on central access to the raw data, and is therefore qualitatively different from ours.

2. *What happens in higher dimensions?* If the covariance matrix is diagonal, parallel invocations of our algorithms
(for each dimension) would work. In other cases, it is not obvious how to scale the binary search approach used here
without incurring penalty exponential in the dimension due to covariance.

[Meta-Review · NeurIPS 2019]

The paper provides new, non-trivial results for scalar Gaussian mean estimation under local differential privacy. The technique in the upper bound is new. Although the lower bound is implied by another recent work, this work is independent. All reviewers have strongly recommended that the authors put less emphasis on the lower bound result and more on the upper bound in the final version, as well as implement the changes mentioned in the rebuttal.